


**Comparing an insurer's perspective on building damages with**
**modelled damages from pan-European winter windstorm event**
**sets: a case study from Zurich, Switzerland**
Christoph Welker[1], Thomas Röösli[2, 3], David N. Bresch[2, 3]
[1] GVZ Gebäudeversicherung Kanton Zürich, Zurich, Switzerland
[2] Institute for Environmental Decisions, ETH Zurich, Zurich, Switzerland
[3] Federal Office of Meteorology and Climatology MeteoSwiss, Zurich, Switzerland
*Correspondence to:* Thomas Röösli (thomas.roeoesli@usys.ethz.ch)
**Abstract**
With access to claims, insurers have a long tradition of being knowledge leaders on damages caused by e.g.
windstorms. However, new opportunities have arisen to better assess the risks of winter windstorms in Europe
through the availability of historic footprints provided by the Windstorm Information Service (Copernicus
WISC). In this study, we compare how modelling of building damages complements claims-based risk
assessment. We describe and use two windstorm risk models: the insurer's proprietary model and the open
source CLIMADA platform. Both use the historic WISC dataset and a purposefully-built, probabilistic hazard
event set of winter windstorms across Europe to model building damages in the canton of Zurich, Switzerland.
These approaches project a considerably lower estimate for the annual average damage (CHF 1.4 million),
compared to claims (CHF 2.3 million), which originates mainly from a different assessment of the return period
of the most damaging historic event Lothar/Martin. Additionally, the probabilistic modelling approach allows
assessing rare events, such as a 250-year return period windstorm causing CHF 75 million damages. Our study
emphasises the importance of complementing a claims-based perspective with a probabilistic risk modelling
approach to better understand windstorm risks. The presented open source model provides a straightforward
entry point for small insurance companies.
**1      Introduction**
Severe windstorms are responsible for widespread socio-economic impacts such as damage to buildings,
structures, transport networks, forests, and even loss of lives. Windstorms represent one of the most damaging
natural hazards in many parts of the world, not least in Switzerland (Imhof, 2011). In the densely populated
canton of Zurich, which is located in north-eastern Switzerland, windstorms are among the most destructive
natural hazards: building damages due to windstorms amount to 30 % of the total amount of building damage
from natural hazard in this region. For comparison, damages due to hailstorms and flooding amount to 41 % and
28 %, respectively (all numbers from 2018; GVZ annual report, 2018; Schadenstatistik VKG, 2020).
In general, the impact of a windstorm in terms of building damages depends on the severity of associated
surface winds and gusts as well as on the exposed values and the respective vulnerability (i.e., damage
susceptibility) of the buildings being subject to the hazard − with both building stock and vulnerability changing
over time. High wind speeds cause large pressure and suction effects, which in turn are responsible for damage
to the roof and the building facade. Damaging winds and violent gusts in the canton of Zurich are mainly due to


the passage of large-scale extratropical cyclones and their associated fronts during autumn and winter as well as
due to mostly local convective storms during summer. Winter windstorms typically cause widespread minor
building damages summing up to large amounts, whereas it is not unusual that summer convective storms cause
major damages of only a few buildings due to locally very high wind speeds.
The cantonal building insurance GVZ compulsorily insures all buildings in the canton of Zurich (with a few
exceptions) against damage due to natural hazards and fire: i.e. in total around 300'000 buildings with a total
sum insured of around Swiss Francs (CHF) 500 billion (in 2018). GVZ is an independent institution of the
canton of Zurich under public law (GVZ homepage, 2020).
Windstorm damage events in the canton of Zurich have been recorded in GVZ's database since 1981. For
example, windstorm Lothar on 26 December 1999 caused total insured building damages of around
CHF 60 million and is by far the most extreme windstorm event in the database. Second largest is windstorm
Burglind on 3 January 2018 (Scherrer et al., 2018), which caused total insured building damages of more than
CHF 14 million. The most extreme summer damage event in GVZ's record is due to a very local, but extremely
intense convective storm on 2 August 2017 with measured maximum gusts of more than 180 km/h in the
lowlands, which caused total insured building damages of approximately CHF 4 million. Even though small-
scale convective storm events are potentially hazardous, in this study we focus on large-scale winter windstorms
only, which were responsible for around three quarters of all insured windstorm damages in the canton of Zurich
since 1981.
Extreme damage events such as those caused by Lothar or even stronger windstorms are rare by definition. For
risk assessment, solid estimates of the probability of occurrence of such events are absolutely essential and
GVZ's claims data of almost 40 years provides a too short observational period. A larger sample of events is
needed for which at least quantitative meteorological data and if possible damage data at ideally high
spatiotemporal resolution are available (e.g., Haas and Pinto, 2012). Observational damage data are generally
sparse and incomplete for historic windstorms in Switzerland (Stucki et al., 2014). Instead, societal actors often
use modelled impacts to manage their risk. Insurance and reinsurance companies apply impact models for their
pricing and governments use modelled risk for option appraisal (e.g., The Economics of Climate Adaptation
Working Group, 2009; Bresch, 2016). Additionally, the information is needed for climate-related financial
disclosure (Surminski et al., 2020). However, only very few impact models are available open source and free
access for users both in the scientific as well as public or private domain.
Typically, risk is modelled as a combination of hazard, vulnerability, and exposure (IPCC, 2014). The hazard
part is the best understood and research culminated in open datasets of historic windstorm events (Roberts et al.,
2014; WISC products, 2019), whereas maximum wind gust speeds are frequently used as hazard component to
assess windstorm risk (e.g., Klawa and Ulbrich, 2003). Vulnerability has been covered by many studies and
reviews (e.g., Della-Marta et al., 2010; Schwierz et al., 2010; Feuerstein et al., 2011; Prahl et al., 2015; Koks
and Haer, 2018). There are many theoretical learnings from these studies, but an implementation in a
comprehensive open source and easy access risk assessment model is still missing. Detailed exposure data are
generally not publicly available and many societal actors have their own detailed view on exposure and do not
need to rely on a publicly available dataset. There are open, spatially explicit datasets available based on the



distribution of nightlight and population (Eberenz et al. 2019), based on the Gross Domestic Product (GDP;
Geiger et al. 2018), or on building data from OpenStreetMap (Koks and Haer, 2018). The sparse availability is
why in some research studies loss ratios were used instead of information on exposure (Donat et al., 2011).
Using the modelling approach for Switzerland, Welker et al. (2016) applied the methods presented first by
Stucki et al. (2015) to a sample of more than 80 high-impact winter windstorms that affected Switzerland in
1871-2011. The approach involves the dynamical downscaling of the Twentieth Century Reanalysis (20CR)
using the Weather Research and Forecasting (WRF) model. The calculated windstorm footprints served as input
for the modelling of economic damages using a precursor of the open source impact model CLIMADA
(CLIMate ADAptation; Aznar-Siguan and Bresch, 2019a). CLIMADA was successfully applied in several other
studies for the purpose of risk assessment and quantification of socio-economic impacts (e.g., Della-
Marta et al., 2010; Schwierz et al., 2010; Raible et al., 2012; Reguero et al., 2014; Gettelman et al, 2018; Walz
and Leckebusch, 2019).
To increase the sample of windstorm footprints available for risk assessment, insurance and reinsurance
companies often combine observed windstorm footprints as far as available with synthetic footprints generated
by stochastic or dynamic atmospheric models. In this way, they obtain a more comprehensive view on risk.
The Windstorm Information Service (WISC) of the Copernicus Climate Change Service aims to provide a
consistent and open database of hazard data to assess the risk of windstorms in Europe for all kinds of players in
the insurance sector and beyond. The centrepiece of the WISC dataset are wind gust footprints at high spatial
resolution of approximately 4.4 km for, on the one hand, a historic hazard event set of around 140 European
winter windstorms in 1940-2014 and, on the other hand, a synthetic hazard event set of around 23'000 events.
Similar to the predecessor project Extreme Windstorms Catalogue (XWS; Roberts et al., 2014), the WISC
historic hazard event set contains windstorms that hit Europe, but provides the corresponding wind gust
footprints at improved spatial resolution and covers more windstorms over a period longer than the data basis
available to most insurance companies. The windstorm hazard event sets as provided by WISC form an
independent database to validate and further develop existing European winter windstorm models. The dataset
can be used for both pan-European analyses and local analyses, as shown in this study.
Using the WISC historic hazard event set allows GVZ in a way to "re-check" historic events. By means of the
synthetic hazard event set, the tail of the hazard and damage distributions should be investigated. However,
Röösli et al. (2018) found that the synthetic hazard event set is not suitable for this purpose. Therefore, we
propose instead a probabilistic windstorm hazard event set based on a method described in
Schwierz et al. (2010) to overcome the shortcomings of the WISC synthetic hazard event set. This new
probabilistic hazard event set of around 4'300 events contains windstorms from the WISC historic hazard event
set altered by various perturbations.
This study shows how GVZ uses both the WISC dataset and the new probabilistic hazard event set for assessing
the potential building damage and risk due to extreme windstorm events (including an evaluation of the
uncertainties of such assessments). A relationship between wind gust speed in the entire area of the canton of
Zurich and associated building damages is found, which allows for a rapid, straightforward estimation of
damage directly after the occurrence of extreme, unprecedented windstorms. This study further shows how GVZ




was able to improve its windstorm risk assessment on the basis of the WISC dataset and the new probabilistic
hazard event set, and could serve as an example for other players in the insurance sector or other societal actors
in Switzerland and in the rest of Europe. At the same time, this study also illustrates selected limitations of the
WISC dataset.

## 117 2 Data and methods

After a description of the insurance claims data (Sect. 2.1) and the windstorm hazard event sets used (Sect. 2.2),
we introduce the GVZ and the CLIMADA risk assessment models applied for damage modelling (Sect. 2.3) and
conclude this section with a brief recapitulation of the risk assessment metrics employed in this study
(Sect. 2.4).

### 122 2.1 Insurance claims data

The windstorm damages of past events are recorded in a proprietary database of GVZ. It consists of almost
40 years of insurance claims data, in total more than 84'000 single wind damage records. From this database all
the events relevant for this study were selected by following the event definition of the windstorm event set
"WISC historic" (Sect. 2.2.1). In total, 18 events are associated with WISC windstorms based on that definition
(see also Table 1). Due to the nature of the database, only the damage reports actually insured by GVZ were
considered. The insurance claims data allow GVZ to assess the risk for its own portfolio by analysing frequency
and severity of past damages, i.e. to assess its risk due to winter windstorm events with a return period smaller
than 40 years. Additional information can help GVZ to put their recorded damages into reference and to get a
better estimate of the risk of events with a return period larger than the 40 years of experience.
For the sake of comparability, the insured damages had to be normalised to present-day exposure levels. In this
study, the applied normalisation considers the general inflation on the basis of the Zurich construction price
index (2020). Hereinafter, both insured and modelled windstorm damages are including occasional deductibles
− so-called "gross damages", to ease comparison.

### 136 2.2 Windstorm hazard event sets

Atmospheric models provide information about winter windstorm events that can be used as hazard component
in a risk assessment model. WISC published several hazard datasets each containing a set of windstorm events
and providing the maximum wind gust per geographic location per event. We used the historic windstorm
footprints (Sect. 2.2.1) and constructed a probabilistic extension based on it (Sect. 2.2.3). In addition, we derived
wind gust footprints from measurements for a selection of present windstorm events (Sect. 2.2.4). The additional
windstorm hazard event sets published by WISC, that are however not considered in this study, are briefly
summarised in Sect. 2.2.2.

### 144 2.2.1 Historic windstorm hazard event set

The historic windstorm hazard event set − denoted "WISC historic" − contains wind gust footprints for around
140 winter windstorm events in Europe in 1940-2014 (i.e., 75 modelled years in total). The events were
selected, on the one hand, based on the high damage they caused and, on the other hand, because of their high


intensity in meteorological terms (i.e., high vorticity). Because of this pan-European perspective, the dataset is
not necessarily specific to windstorms in the canton of Zurich. Nevertheless, the high-impact windstorms
Lothar/Martin (26–28 December 1999) and other intense windstorms such as Vivian/Wiebke (26 February–
1 March 1990) are included.
The windstorm footprints were computed by running the UK Met Office Unified Model (MetUM; Davies et al.,
2005) at approximately 4.4 km resolution with ERA-20C reanalysis (Poli et al., 2016) and ERA-Interim
reanalysis (Dee et al., 2011) as boundary conditions, covering Europe and parts of the North Atlantic. ERA-20C
was used for all windstorm events in 1940-1979 and ERA-Interim for all events in 1979-2014.
Each of the footprints is composed of gridded maximum 3-second gusts, with maxima determined for a 72-hour
time window. This relatively long time window was chosen, because it is widely used in the insurance sector
(WISC products, 2019). However, it also implies that the footprints of directly successive events (i.e., with a
time difference of less than 72 hours) such as Lothar (26 December 1999) and Martin (27-28 December 1999)
are combinations of the footprints of both successive events. In this study, the WISC windstorm footprints for
events that have overlapping time windows are combined to represent one event – as insurance claims data does
often not represent the exact time/date of damage either (for various reasons, a key one being reporting
uncertainties). This combination is necessary to make sure that a maximum that occurred only once (e.g., the
wind gusts reached during Lothar) is only represented once in the hazard event set (as event Lothar/Martin) and
is not represented twice (once as Lothar and once as Martin). There are five pairs of windstorms with
overlapping time windows in the original dataset that were combined by taking the maximum wind gust of both
footprints at each location, giving in total 142 windstorm events (Table 1). The problem of overlapping
windstorm footprints and the resulting combination of events could have been prevented by incorporating the
geographical information into the event definition. For example, Roberts et al. (2014) aggregated only the wind
gusts within a certain radius around the windstorm centre into a footprint to avoid this problem.
The wind gust speeds from "WISC historic" are considered to be realistic compared to observations for areas at
sea level (WISC products, 2019). However, with regard to the hilly topography of the canton of Zurich the
question arises as to how realistic the underlying model topography is in comparison to the real topography and,
as a result, how good the height-dependent wind gust speeds are compared to observational data. Even though
this could not be finally clarified in this study since available wind measurements are generally too sparse for
historic windstorms in the canton of Zurich, a correction of all the WISC wind gusts in the form of simple
correction factors does not seem reasonable and was therefore not applied.
**2.2.2    Other WISC hazard event sets**
There are two additional windstorm hazard event sets published by WISC, that are however not analysed in
detail in this study:
1.    The operational windstorm hazard event set − denoted "WISC operational" − contains around
110 windstorm events in 1979-2017 and thus more recent events than the windstorm hazard event set "WISC
historic" used in this study, which contains windstorm events until 2014 only. "WISC operational" is based on a
new generation of atmospheric reanalysis, the ERA5 reanalysis (Hersbach and Dee, 2016). As it does not cover




the time range 1940-1979 (compared to "WISC historic") it does not complement the recorded damages by
providing information about historic events not covered by GVZ's claims database.
2.         The synthetic windstorm hazard event set − denoted "WISC synthetic" − was created within the
UPSCALE (UK on PRACE - weather-resolving Simulations of Climate for globAL Environmental risk;
UPSCALE, 2020) modelling framework and is a physically realistic set of plausible winter windstorm events in
the period 1985-2011 based on the climatic conditions of that period. The modelling framework developed five
ensembles. The dataset contains wind gust footprints for around 23'000 synthetic windstorms: i.e., three sets of
7'660 events each. Each of the three sets covers 135 modelled years. The original idea of the hazard event set
"WISC synthetic" was to use wind information from climate models to provide wind gust footprints for winter
windstorms in Europe with a return period of 250 years or even higher. However, this hazard event set was not
considered because the findings of Röösli et al. (2018) could be replicated in this study, showing that the dataset
does not contain the maximum wind gust speeds we would expect from the distribution of the historic
windstorm hazard events (Fig. A1) nor the high intensities we would expect from very rare, high-impact
windstorm events (Fig. 1).
For a detailed description of all unused windstorm hazard events sets provided by WISC, we refer to the
documentations available online at WISC products (2019) and WISC hazard event set description (2019).

### 2.2.3    Probabilistic windstorm hazard extension

Based on "WISC historic", we generated an additional probabilistic windstorm hazard event set − denoted
"WISC probabilistic extension". By applying a method described in Schwierz et al. (2010), the individual
windstorm events in "WISC historic" (parent events) were altered to create 29 altered offspring events by
various perturbations: e.g., spatial displacement and by weakening / intensifying the wind speeds (non-altered
wind speeds are spatially displaced only). The spatial displacement was undertaken by shifting the respective
windstorm footprint by about 20 km to the north, south, west, or east. The wind gust speeds were intensified and
weakened by no more than 3 m/s (normally much less) according to the probabilistic alteration of wind speeds
in Eq. (1), with a scale parameter $\alpha = 0.0225$ and a power parameter $\beta = 1.15$:

$$windspeed_{scenario\ 1} = windspeed_{original} + \alpha * windspeed_{original}{}^{\beta}$$

$$windspeed_{scenario\ 2} = windspeed_{original} - \alpha * windspeed_{original}{}^{\beta}$$

$$windspeed_{scenario\ 3} = windspeed_{original} + \alpha * \sqrt[\beta]{windspeed_{original}}$$

$$windspeed_{scenario\ 4} = windspeed_{original} - \alpha * \sqrt[\beta]{windspeed_{original}}$$
(1)

$$windspeed_{scenario\ 5} = windspeed_{original} + \frac{\alpha}{2} * windspeed_{original}{}^{\beta}$$
$$+ \frac{\alpha}{2} * \sqrt[\beta]{windspeed_{original}}$$




These newly created "probabilistic" footprints can be viewed as scenarios of plausible windstorms as they only
differ slightly from historic events, retaining both the spatial extent and general structure.
For using the scenarios in a qualitative risk assessment framework, the probabilistic windstorm footprints can be
used as they are, but for a quantitative risk assessment the frequencies of the windstorm footprints need to be
estimated. In an effort to assign reasonable frequency estimates to the probabilistic windstorm footprints, we
considered the distribution of the historic, pan-European Storm Severity Index (SSI; e.g., Lamb and Frydendahl,
1991; Leckebusch et al., 2008; Dawkins et al., 2016). Similar as in Schwierz et al. (2010), the algorithm of
creating the probabilistic windstorm footprints was configured to recreate the cumulative distribution function of
a generalized extreme value (GEV) distribution fitted to the historic SSI values. We defined the frequency of all
probabilistic windstorm footprints to be equal and to sum up to the frequency of the parent windstorm. We then
selected a set of parameters for weakening and intensifying the wind speeds (parameters $\alpha$ and $\beta$ in Eq. (1)) that
resulted in a similar probabilistic distribution of SSI as the extrapolated distribution from the historic SSI values.
For the probabilistic hazard event set to best represent the tail of the historic distribution, we determined a
combination of $\alpha$ and $\beta$, that minimises the difference in the cumulative distribution functions for events that are
rarer than 75 years.
"WISC probabilistic extension" includes footprints for 4'118 probabilistic windstorm events, along with the
142 original windstorm events in "WISC historic" (Table 1), and provides a basis of an event-based risk
assessment for winter windstorms with return periods of around 250 years, a scenario relevant for regulatory
requirements in the insurance sector. It is important to note that this method incorporates a lot of uncertainty,
including but not limited to the sampling uncertainty of rare events in a relatively short time range (i.e., 75 years
in case of "WISC historic").
Encouragingly, the hazard event set "WISC probabilistic extension" shows considerably higher wind gust
speeds in the canton of Zurich as compared with "WISC synthetic" (Fig. 1). Nonetheless, the maximum wind
gust speeds of the most extreme event in "WISC probabilistic extension" are not considerably higher than those
of Lothar/Martin, the most extreme event in both "WISC historic" and the insurance claims data.

### 235   2.2.4   Observed footprints for current windstorms

Real-time wind gust observations can serve as the hazard part of the damage model for a rapid damage
estimation directly after the occurrence of an extreme windstorm event. Such "observed" windstorm footprints
can also be used for further validation of GVZ's damage modelling approach (Sect. 2.3). To create such
footprints, we used interpolated wind gust measurements in the canton of Zurich based on the Common
Information Platform for Natural Hazards (GIN; GIN platform, 2019) for a selection of seven winter
windstorms in the years 2017 and 2018. With the exception of winter windstorm Burglind hitting Switzerland
on 3 January 2018, the windstorms considered caused only minor damages in the canton of Zurich. The
individual windstorm footprints are based on a total of around 110 measurement stations in the canton of Zurich
and in the immediate vicinity (i.e., buffer zone with a distance of 20 km around the polyline of the canton). For
spatial interpolation, we applied an Inverse Distance Weighting (IDW) interpolation with the Shepard method
used for weight calculation. In this study, the gridded wind gust footprints derived from measurements have a
horizontal resolution of 2 km. The topography of the canton of Zurich is not considered in the applied





interpolation method and unquestionably the quality of the derived windstorm footprints could be improved by
using a more elaborate interpolation method, which takes account of the topography.

## 2.3 Damage modelling approaches

The windstorm footprints of the different hazard event sets described in the previous section were used as input
for damage modelling and GVZ's proprietary windstorm damage model was applied for this (Sect. 2.3.1). In
addition, the CLIMADA impact model was used to be able to publish the method used in this study with open
data and open source code (Sect. 2.3.2).
In both damage models, the extent of damage results from the intensity of the windstorm event (i.e., hazard), the
value of the asset (i.e., exposure), and the susceptibility of the asset to damage (i.e., vulnerability). This concept
is broadly used and is explained in more detail in Aznar-Siguan and Bresch (2019a). In this study, the
windstorm hazard assessment is based on the winter windstorm footprints described in Sect. 2.2. The exposure
is the value of the buildings in the canton of Zurich and the vulnerability is described by a functional
relationship that defines how much the buildings are damaged at a certain wind gust speed. In both damage
models, we use the vulnerability curve of Schwierz et al. (2010). This vulnerability curve combines the damage
degree and the percentage of assets affected. Only damage to buildings is estimated. The estimate does not
include damage to movable property, damage to infrastructure, nor business interruption.

### 2.3.1 GVZ damage model

The damage estimates in this model are computed using a rather conventional modelling framework and the
reduced complexity of the approach allows a well interpretable assessment of the model skill. Normally, GVZ
uses its damage model directly after the occurrence of a windstorm event to estimate the expected building
damage. Furthermore, GVZ applies the damage model to estimate the damage potential and the risk associated
with windstorms with regard to solvency considerations and prevention options. The main points of the
modelling approach are described in the following.
The initial step is a simple spatial overlay of the gridded maximum wind gust speeds during the respective
windstorm event with GVZ's current building stock (from 2018; without sublevel garages, as they are usually
not affected by windstorms), where GVZ's proprietary building database with information about e.g. the sum
insured of each building and the publicly available building footprints (GIS browser Zurich, 2019) were used.
GVZ's insurance penetration in the canton of Zurich is almost 100 %. In the damage model, damage is possible
from a wind gust speed of more than 90 km/h, and only buildings affected by such gusts were considered in the
following modelling steps.
Figure A2 shows the spatial distribution of all insured buildings in the canton of Zurich as well as of the total
sum insured at municipal level. The aggregated sum insured for all buildings in the two main cities, Zurich and
Winterthur (municipal boundaries indicated by blue polygons), accounts for almost 40 % of the total insured
value for the entire canton.
To estimate the damage in monetary terms, the value of each individual building (i.e., its insured value) was
multiplied by the factor "Mean Damage Degree" (MDD) according to Schwierz et al. (2010), where the gust



speeds at building level computed in the first step were converted into the corresponding MDD factors. The
MDD factors are a non-linear function of the maximum wind gust speed during a windstorm event. The same
vulnerability curve of Schwierz et al. (2010) is also implemented in the open source impact model CLIMADA
(Aznar-Siguan and Bresch, 2019a). The vulnerability curve is diagrammed in Welker et al. (2016).
In the next step of the damage model, the probability of buildings affected is calculated with a stochastic
approach. The respective windstorm event was automatically categorised according to its severity (here,
according to the 95th percentile of all gust speeds at building level in the canton of Zurich), from which the
assumed degree of impact is derived. The degree of impact for the different windstorm categories (a percentage
of total affected buildings for the canton of Zurich, $m$) was derived from proprietary event damage data from
GVZ's database. Then, a random sample of $m$ buildings was selected, with the number $m$ depending on the
windstorm's severity. Only buildings with MDD > 0 were considered, i.e. only those buildings with potential
damage > 0. For the selected buildings, the amount of damage at building level was summed to obtain the total
damage for the entire canton. This procedure of random sampling was repeated 1'000 times giving a total
damage range for each windstorm event. Unless otherwise stated, for each windstorm the median of the damage
distribution is given hereinafter.
### 2.3.2    CLIMADA impact model
The windstorm damage model in the open source risk assessment platform CLIMADA is relying on open data
only and that is why it is deviating in some aspects from GVZ's approach described above. As the windstorm
hazard component is open, it is identical to the hazard input used in case of the GVZ damage model. The
exposure is based on public data instead of GVZ's proprietary portfolio information. CLIMADA uses produced
capital for Switzerland published by the World Bank (2018) as the total value of physical assets for Switzerland
and further uses a combination of nightlight intensity and population density to create a reliable geographical
distribution of the assets (Eberenz et al., 2019). The resulting values are then distributed to building footprints
from OpenStreetMap (OpenStreetMap contributors, 2017). Analogous to the GVZ damage model, CLIMADA
uses the MDD curve of Schwierz et al. (2010). Instead of a random resampling of affected buildings, the MDD
factor is combined with the deterministic factor "Percentage of Assets Affected" (PAA).
As the total value of the exposure is different between the GVZ exposure, the CLIMADA exposure, and the
exposure used in Schwierz et al. (2010), the MDD and PAA factors might be wrongly scaled for this study. In
the CLIMADA model setup used, we adjusted for this by linearly scaling the MDD and PAA factors to reduce
the difference of the modelled damages and the insured damages for matching events (i.e., by minimising the
root-mean-square deviation, RMSD). This adjustment conserved the shape of the original vulnerability curve.
The CLIMADA impact model and the GVZ damage model have a different sensitivity to the hazard intensity: in
CLIMADA, damage is possible for a wind gust speed of 72 km/h (20 m/s) and above, in the GVZ damage
model for 90 km/h (25 m/s) and above.
### 2.4    Assessment of potential windstorm damage and risk
Risk is defined here as the product of the extent of damage and the probability of damage. The probability of
damage is driven, on the one hand, by the probability that the building is within the area of high wind gust


speeds and, on the other hand, by the return period of the windstorm event. The probability, that the building is
within the area of high wind gust speeds is incorporated in the modelled damage amount by the spatially explicit
modelling approach and the vulnerability, which includes the percentage of assets affected (in case of
CLIMADA). The return period or frequency of windstorm events is derived from the hazard event sets. Return
periods express the probability of occurrence of windstorm events (e.g., an event with a return period of 250
years is expected on average every 250 years).
There are several risk assessment metrics that can be calculated with a set of event damages, which are the main
result from the damage modelling described above.

### 2.4.1    Average annual damage

The average annual damage (AAD) is an important risk measure in the insurance industry. It describes the risk
from all events reported on an annual basis:

$$AAD = \frac{sum\ of\ all\ event\ damages}{time\ range\ covered\ by\ event\ set} = \sum_{event\ i} event\ damage_i * annual\ frequency_i \qquad (2)$$

### 2.4.2    Exceedance frequency curve

Using the annual frequencies of the events in a hazard event set, it is possible to determine at what frequency a
certain damage amount is exceeded. The largest damage amount is exceeded once in the time range covered by
the damage event set, the second largest damage amount is exceeded twice, the third one thrice and so on. The
exceedance frequency curve shows the damage amount as a function of exceedance frequency. For large
damage amounts, this matching typically relies on only a few damage events, which increases the statistical
uncertainty.

### 2.4.3    Pareto pricing

In the insurance industry, the concept of "Pareto pricing" is a simple approach to represent and extrapolate the
distribution of a damage event set to define the price of insurance contracts (Mitchell-Wallace et al., 2017). We
imitated this pricing method by fitting a Generalized Pareto Distribution (GPD) to damage event sets using a
Maximum Likelihood Estimate (MLE). We do this even though some assumptions in statistical theory are not
valid for these datasets (e.g., windstorm damage event sets are clustered which breaks the independence
assumption), as we use the GPD only to show the underlying sampling uncertainty. To fit a GPD to a damage
event set, only the threshold has to be chosen. We chose a threshold for each damage event set, which results in
a parameterised GPD with similar exceedance frequencies for the largest damage amount in the event set. For
the insured damages we chose a threshold of CHF 0.4 million and for the modelled damage event set based on
"WISC historic" we chose a threshold of CHF 0.1 million. By using the percent point function (the inverse of a
cumulative distribution function) on the fitted distributions, an exceedance frequency curve for the fitted
distribution was calculated.
To illustrate the statistical uncertainty of the exceedance frequency curve, we undertook a resampling. In the
resampling, we generated 200 random samples from the fitted distribution and used the MLE to fit a GPD to
each random sample. The exceedance frequency curves of these resampled distributions illustrate the





uncertainty especially for rare events with a high return period. We show the 90-% confidence interval of
damage amounts for each exceedance frequency, which spans from the 5th percentile to the 95th percentile of
the 200 samples.
**3      Results**
**3.1     Single events**
The damage due to Lothar/Martin is by far the largest windstorm event damage in GVZ's insurance claims
database (Fig. A3a): Lothar/Martin caused insured damages of CHF 62.4 million. Lothar/Martin is the most
damaging windstorm event in the canton of Zurich in both the 34-years period of insurance claims data as well
as in the 75-years period of "WISC historic". The damages modelled with the GVZ damage model range
between CHF 58.0 million and CHF 69.0 million, and the median of all modelled damages amounts to
CHF 62.7 million (Fig. A3b). For Burglind, the most damaging event of the "observed footprints", the modelled
damages range between CHF 10.4 million and CHF 14.5 million, with a median of CHF 12.0 million. For
comparison, the insured damages amount to CHF 14.2 million. Thus, damages associated with intense
windstorm events like Lothar/Martin or Burglind are very well modelled with GVZ's damage modelling
approach, providing confidence in the methodology. For all recorded windstorm events since 1981 (including
the additional seven windstorms in 2017 and 2018), the RMSD between the insured damage and the median
modelled damage amounts to CHF 2.4 million. Furthermore, the example of Burglind shows that our
methodology of creating windstorm footprints on the basis of interpolated wind gust observations (Sect. 2.2.4) is
suitable for present and probably also for future windstorm events.
**3.2     Average annual damage**
The average annual damage (AAD) calculated based on the insured damages (i.e., the mean damage over the
observational period of 34 years) is almost twice as high as the AAD computed on the basis of "WISC historic"
(Table 2). Several factors contribute to the fact that the AAD is higher for the insured damages than for the
modelled damages based on "WISC historic": (i) the occurrence of the very intense event Lothar/Martin, along
with other intense events, in the relatively short available period of insurance claims data (Figure A3a), (ii) the
higher damages of events in the 5-year return period range (Table 2), and (iii) the different number of events per
year considered. The hazard event set "WISC probabilistic extension" was created to best represent the low-
frequency tail of the pan-European SSI and not the full distribution of (high frequency) damages in the canton of
Zurich. Nevertheless, the modelled AAD based on the GVZ damage model and "WISC probabilistic extension"
is close to the AAD of "WISC historic".
**3.3     Assessment of risks due to extreme windstorm events**
Figure 2 shows GVZ's windstorm risk assessment of building damage, including uncertainty, on the basis of all
available data sources. Based on the insurance claims data only, the return period for the extreme windstorm
event Lothar/Martin is estimated to be 34 years (blue squares). Based on "WISC historic", the return period for
Lothar/Martin is estimated to be 75 years (yellow dots). Based on the hazard event set "WISC probabilistic


extension" and using GVZ's approach for damage modelling, the return period for a damage amount due to
Lothar/Martin would be around 125 years (red diamonds).
The extrapolated event damage with a return period of 250 years amounts to about CHF 500 million for
"WISC historic" and using the same method for the insured damages the extrapolated 250-year event damage
would be even higher, around CHF 2.4 billion (yellow and blue lines in Fig. 2). Contrary to this, the 250-year
event damage amounts to only about CHF 75 million in case of the hazard event set "WISC probabilistic
extension" (red diamonds). The 90-% confidence interval, which represents the sampling uncertainty of the
extrapolation of the damage exceedance frequency, based on "WISC historic" provides a range for the 250-year
return period damage of CHF 19 million to CHF 33 billion (yellow ribbon). As "WISC probabilistic extension"
is based on the same historic information this uncertainty also applies to its results.
Interesting to see in Fig. 2 is that the tail of the modelled damages on the basis of "WISC probabilistic
extension" is reaching far smaller damages per return period than the two extrapolations based on the fitted
distributions. Evident "jumps" in the modelled damage (e.g., at return periods of approximately 30 years,
70 years, and 90 years) result from the discrete categorisation of the individual windstorm events and the
assumed degrees of impact, respectively, as applied in GVZ's damage modelling approach (Sect. 2.3.1).

### 3.4    Reproducibility of the results using CLIMADA

In general, GVZ's proprietary windstorm damage model is suitable for correctly simulating building damage in
the canton of Zurich (see Fig. 3, Fig. A3, and Sect. 3.1). Using the calibrated CLIMADA impact model for
windstorm damage modelling is also suitable and the corresponding RMSD amounts to CHF 1.5 million for all
recorded windstorm events since 1981 for which WISC wind gust footprints are available (excluding the
additional windstorms in 2017 and 2018). The statistics in Table 2 calculated using the GVZ damage model
were also calculated using the CLIMADA impact model and the results can be found in Table A1. In summary,
it can be stated that the setup of the two damage models applied works well and e.g. replicates the order of the
events, provides a reasonable modelled damage for historic events (compared to insurance claims data), and
both RMSD are sufficiently good.
The exceedance frequency curve of the modelled damages based on "WISC probabilistic extension" and the
CLIMADA impact model (green triangles in Fig. 2) show in general lower values compared to the damage
modelling using the GVZ approach (red diamonds), in particular for return periods between 30 and 70 years.
This difference is also reflected in the scatter plots in Fig. 3, where in Fig. 3a the GVZ damage model shows an
overestimation of the damage amount due to the windstorm event Vivian/Wiebke (with insured damage of
approximately CHF 11 million), whereas the CLIMADA impact model shows an underestimation for the same
event. The reason for this over- and underestimation of the damage in case of events such as Vivian/Wiebke
could be due to the hazard or exposure part of the respective model, but is more likely due to the applied
vulnerability curve itself. Apparently, the two damage models perform differently for windstorm events in a
medium intensity category. This difference between the two models also becomes evident regarding the AAD
risk metric: the AAD of the CLIMADA impact model with "WISC historic" amounts to CHF 1.1 million
(Table A1) and is thus almost a third smaller than the AAD associated with the GVZ damage model
(CHF 1.4 million). In addition, the curve of the modelled damages is much smoother in case of CLIMADA




(Fig. 2), which can be explained by the fact that in CLIMADA the smooth curve of the PAA factors is used.
This shows the importance of the applied vulnerability curve in the presented damage modelling approach.
**3.5     Rapid damage estimation**
Rapid damage estimation directly after a windstorm event is very useful for insurance companies to get a first
rapid assessment of the damage to be expected and to e.g. assign their staff accordingly. For current windstorm
events, the GVZ does this using its damage model and the wind gust footprints based on "observed footprints"
(Sect. 2.2.4). The 95th percentile of the wind gust speeds at building level in the entire area of the canton of
Zurich, which is also used in GVZ's damage model to categorise windstorm events (Sect. 2.3.1), is used as a
rapid indicator of the range of possible damages. This process is illustrated in Fig. 4. With the help of the dataset
"WISC probabilistic extension", assessments can also be made about potential damages from unprecedented,
extreme windstorm events. The uncertainty of the damage assessment for such extreme events can be visualised
by the large number of available (extreme) events. In total, "WISC probabilistic extension" contains 17 events
which are potentially more damaging than Lothar/Martin. A (modelled) total damage amount of more than
CHF 96 million is associated with the most extreme windstorm event in "WISC probabilistic extension"
(Fig. 1). Thus, this windstorm is potentially about 1.5 times as damaging as Lothar/Martin.
Figure 4 further shows, by the length of the red bars, the stochastic component in GVZ's damage modelling
approach, which tries to approximate the random selection as not every building is equally affected during a
windstorm event (Sect. 2.3.1). The range of modelled damages (length of red bars) increases with increasing
wind gust speed. On the other hand, the quotient of the range of modelled damages and the median of the
damage distribution (red points) generally decreases with increasing wind gust speed. "Jumps" in the modelled
damage (e.g., for wind gust speeds lower than 126 km/h) again result from the discrete categorisation of the
individual windstorm events in the GVZ damage model.
The absolute difference between the modelled damage amount and the corresponding value of the regressed
relationship (red points and solid red line in Fig. 4) generally increases with increasing wind gust speed.
Accordingly, the number of available wind gust footprints decreases with increasing wind gust speed.
**4        Discussion**
Any information about the historic risk of winter windstorms in the canton of Zurich contains the record of the
event Lothar/Martin. As this is the most damaging event in the record by far, the general risk assessment is
connected to the assessment of the return period of such an event damage, which will always be uncertain. We
argue that the return period based on the historic windstorm footprints (75 years) is much more reliable than the
return period based on the insured damage record (34 years). Other information, like the return period of
Lothar/Martin's damage amount based on "WISC probabilistic extension" and an independent catalogue of
historic windstorms in Switzerland by Stucki et al. (2014) suggest that the return period of such a damage
amount could be even rarer than 75 years. This clearly shows the added value that GVZ achieves in its risk
assessment through applying the WISC wind data compared to using insurance claims data only − and, above
all, through the additional dataset "WISC probabilistic extension". The return period of extreme windstorm
events such as Lothar/Martin can now be assessed more reliably.





The windstorms Lothar and Martin affected, in addition to Switzerland, in particular France, Belgium,
Luxembourg, and Germany. The original industry damage associated with Lothar and Martin amount to
approximately EUR 5.8 billion and EUR 2.5 billion, respectively (PERILS, 2020). The return period for
exceeding the damage amount due to Lothar alone in all of Europe was estimated to be 15 years by
Munich Re (2002) and the return period for the cluster of the three windstorms in December 1999 Anatol
(3 December 1999), Lothar, and Martin was estimated to be between 22 and 45 years
(Renggli and Zimmerli, 2016). This study shows that it is important to make a distinction between the return
period of an event like Lothar/Martin in all of Europe and the return period of this event locally, in a relatively
small region. The damage modelling shown in this study, using the event set "WISC historic" and the local
exposure information, enables a much more reliable derivation of the return period specific to GVZ than the
existing scientific work is able to provide.
Based on "WISC historic" and the GVZ damage model, the average annual damage for building damages in the
canton of Zurich amounts to CHF 1.4 million according to our calculation and we argue that this is the best
available estimate for the AAD. However, this estimation is still uncertain due to the high sampling uncertainty,
the uncertainty associated with the assessment of the event Lothar/Martin, and the uncertainty with regard to the
damage modelling itself. For comparison, in the last 10 years GVZ has experienced yearly damage from all
natural hazards of CHF 16 million and additionally yearly damage by fire of CHF 42 million (all numbers from
2018; GVZ annual report, 2018). Compared to the risk from these hazards, the estimated AAD from winter
windstorms of CHF 1.4 million is relatively small. However, the occurrence of windstorm events such as
Vivian/Wiebke, Lothar/Martin, and Burglind has shown that single windstorms are able to cause huge damage
amounts and they are consequently an important causal element when assessing capital requirements.
Insurance companies undertake their business under a strict regulatory environment, and having enough capital
to cover rare events is one of the regulatory requirements. The damage amount reached on average every
250 years is an often-mentioned indicator for such a rare event. However, the insured damages and also the
modelled damages based on "WISC historic" do not span a long enough period by far to make an empirical
prediction of a damage amount with a return period of 250 years. All methods of extrapolation from these
datasets suffer from the sampling uncertainty (shown as confidence intervals in Fig. 2). The hazard event set
"WISC probabilistic extension" uses the distribution of pan-European SSI values to create a set of probable
events with higher return periods than "WISC historic". The uncertainty of the return periods of such events
however cannot considerably be reduced compared to "WISC historic", because it relies on the same historic
information. Despite the uncertainty, it can nevertheless be important to study the sensitivity of the 250-year
return period damage to changes in the portfolio (like growth or changed building codes), changes in the
deductible or other changes. "WISC probabilistic extension" provides windstorm footprints of events with a
return period of 250 years (and more), that allow the modelling of damages with changes in the exposure or the
vulnerability.
It comes as no surprise that the choice of the vulnerability curve in the damage modelling approach applied
strongly influences the results of the damage estimation (e.g., Koks and Haer, 2018), and unsurprisingly no
optimal "one-size-fits-all" vulnerability curve exists. Every damage model behaves differently, not least because
different vulnerability curves are used and each of the damage models has been calibrated differently. The
vulnerability curve of Schwierz et al. (2010) is based on movable property and building damages associated
with European winter windstorms. The rather general function does not make a distinction between building
types, in contrast to other available functions (e.g., Feuerstein et al., 2011). For a modelling setup with focus on
the hazard, the vulnerability curve of Schwierz et al. (2010) is however suitable and was successfully applied in
earlier studies (e.g., Stucki et al., 2015; Welker et al., 2016). The function does not require detailed information
regarding the values at risk, which is certainly an advantage for such insurance and reinsurance companies that
do not have detailed exposure data for their damage modelling. The vulnerability assumed in this study and the
corresponding hazard intensity only considers the maximum gust speeds during an event and not the duration of
high wind gusts within a windstorm event, which can however have a major impact on the damage to be
expected. Taking the windstorm duration into account (e.g., Etienne and Beniston, 2012) could improve our
damage modelling, and it is planned to implement this in a future version of GVZ's damage model.
Furthermore, it is not considered that buildings are partially adapted to local wind conditions (e.g., multi-storey
buildings or exposed buildings located on mountain tops).
Not every building is equally affected during a windstorm event. To take that into account, in the GVZ damage
model a random resampling of affected buildings was applied according to an assumed degree of impact (red
bars in Fig. 4). The assumed degree of impact was derived according to the respective severity category of the
windstorm. This severity categorisation and the assumed degrees of impact are inevitably relatively rough in
GVZ's current model setup, because the assumptions are based on insurance claims data from only a few past
windstorm events in the canton of Zurich. With every further windstorm, these assumptions will however
become more reliable in the future. In contrast, the deterministic PAA values (Schwierz et al. 2010), as used in
the CLIMADA impact model, are much smoother and thus allow a smooth damage modelling (Fig. 2).
However, these values are not specific for windstorms in the canton of Zurich and they do not allow a stochastic
sampling as in GVZ's damage modelling approach.
The rapid estimate of the damage potential in the event of extreme, unprecedented windstorm events shown in
Fig. 4 is just one example of how the WISC data and in particular the additional damage event set
"WISC probabilistic extension" can be used for insurance applications. The idea was to be able to make a
statement about the damage to be expected simply based on available wind observations in the area of the
canton of Zurich. It is always important for insurance companies to be able to give a damage assessment as
rapidly as possible after an event, not least when it comes to media inquiries. However, one should keep in mind
that the uncertainty shown does not incorporate the full uncertainty of the damage estimate, but rather the
uncertainty that results from the random selection as not all buildings are affected equally during a windstorm
event. In a future study, it would be interesting to quantify the full uncertainty of the rapid damage estimate.
Not least, the WISC wind data enable insurance companies to evaluate the variability and long-term changes of
winter windstorms and their associated damage since 1940. Besides a marked interannual and decadal-scale
variability of windstorms in the canton of Zurich, we find a tendency for more intense windstorms since
approximately mid of the 1980s (Fig. A3d). One possible reason for this positive trend is that "WISC historic"
consists of two "parts" with different databases: until 1979, the ERA-20C reanalysis (Poli et al., 2016) was used
for downscaling, followed by the ERA-Interim reanalysis (Dee et al., 2011). Furthermore, a change in the large-
scale, atmospheric dynamics has been observed in recent decades, which was conducive to increased winter


windstorm activity and intensity in Switzerland (Welker and Martius, 2015). This change was accompanied by
an atmospheric circulation pattern resembling a southeastwardly displaced winter North Atlantic Oscillation
(NAO) pattern. Which of the two reasons is dominant for the found positive tendency in winter windstorm
intensity and associated damages in the canton of Zurich could not be finally clarified in the present study.
Furthermore, how winter windstorm activity and intensity in mid-latitude Europe will change in a future warmer
climate is still uncertain (Catto et al., 2019).
**5      Conclusion**
This study is an example of how a regional building insurance company in Switzerland uses the open database
of European windstorm event sets provided by WISC in combination with a probabilistic extension for their
assessment of potential building damages and risks as a result of extreme winter windstorm events, including an
evaluation of the uncertainties. The windstorm event Lothar/Martin in December 1999 is the most damaging
event in both the insurance claims data and "WISC historic" (damage of more than CHF 60 million). The
average annual damage for building damages in the canton of Zurich is CHF 1.4 million, computed based on
"WISC historic" and the GVZ damage model.
Both the insurance claims data and the modelled building damages based on "WISC historic" are rather
unsuitable for evaluating rare windstorm damage events with return periods considerably exceeding the
observational period. The new hazard event set "WISC probabilistic extension" projects a damage amount of
approximately CHF 75 million for a return period of 250 years, while the uncertainty for an extrapolation to
such return periods is still very large. However, the probabilistic hazard event set allows for testing the
sensitivity of the risk to e.g. changes in the insurance portfolio or in the insurance condition (e.g., the deductible)
for events of a higher intensity than the observed historic events.
Our analysis is implemented in GVZ's proprietary windstorm damage model as well as in the open source risk
assessment platform CLIMADA (Bresch and Aznar-Siguan, 2019a). This guarantees scientific reproducibility
and offers insurance companies and other societal actors in Switzerland and the rest of Europe the opportunity to
apply the shown methodology to their own portfolio with a low entry threshold. This study illustrates how open
climatological data and open source damage models can be used to assess windstorm risks in Europe and how
this approach complements risk assessments based on proprietary insurance claims data only.
There is a growing societal need for physical risk disclosure, not least in the context of the Task Force for
Climate-related Financial Disclosure (TCFD; Surminski et al., 2020). The presented methodology, in particular
the combination of the WISC hazard data with the open source CLIMADA platform, can be used for such a
disclosure report.
**Code availability and data availability**
The scripts reproducing the main results of the paper and the figures is available under
https://github.com/CLIMADA-project/climada_papers. The probabilistic hazard event set "WISC probabilistic


extension" for each European country is made available for download under https://doi.org/10.3929/ethz-b-
000406567 (Röösli and Bresch, 2020).
CLIMADA is openly available at GitHub (https://github.com/CLIMADA-project/climada_python, Bresch and
Aznar-Siguan, 2019a) under the GNU GPL license (GNU operating system, 2007). The documentation is hosted
on Read the Docs (https://climada-python.readthedocs.io/en/stable/, Aznar-Siguan and Bresch, 2019b) and
includes a link to the interactive tutorial of CLIMADA. CLIMADA v1.4.1 was used for this publication, which
is permanently available at the ETH Data Archive: http://doi.org/10.5905/ethz-1007-252 (Bresch et al., 2020).

**Author contribution**
CW and TR share first co-authorship and contributed equally to defining the case study, performing the
analyses, writing the article, and participating in the review process. CW developed the GVZ damage model and
TR generated the hazard event set "WISC probabilistic extension". DNB contributed to writing the article,
conceptualised CLIMADA, and over-saw its implementation in Python, based on the previous MATLAB
implementation by himself.
**Competing interests**
The authors declare that they have no conflict of interest.
**Acknowledgements**
We are very thankful to the WISC consortium and project team for making all the data and documentation
available and fully open access. Map data copyrighted OpenStreetMap contributors and available from
https://www.openstreetmap.org. We want to thank Jan Hartman for his help implementing the Storm Europe
hazard module in the Python version of CLIMADA, Evelyn Mühlhofer for implementing the OpenStreetMap
exposure module in CLIMADA, as well as Samuel Eberenz and Maurice Skelton for providing valuable input
on the manuscript.

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

wohnen/wohnbaupreise/zuercher-index-der-wohnbaupreise.html, last access: 14 January 2020.





**Table 1:** Summary statistics for the windstorm hazard event sets and insurance claims data used in this study.

| Dataset | Available years (period) | Total number of available windstorm hazard events | Number of damage events in the canton of Zurich |
|---|---|---|---|
| "WISC historic" | 75 (1940-2014) | 142 | 27 |
| "WISC probabilistic extension" | 2'250 (30*75) | 142 (parent events) and 4'118 (altered offspring events) | 754 |
| "WISC synthetic" | 405 (3*135) | 22'980 | 42 |
| "WISC operational" | 39 (1979-2017) | 106 | untested |
| "Observed footprints" | 2 (2017-2018) | 7 | 7 |
| Insurance claims data | 36 (1981-2014 and 2017-2018) | - | 18 ("WISC historic") and 7 ("observed footprints") |

**Table 2:** Annual average damage (AAD) and event damage for different return periods (RP) and the windstorm
event Lothar/Martin on the basis of insurance claims data and modelled damages using the GVZ damage model
and the hazard event sets "WISC historic" and "WISC probabilistic extension", respectively.

| | Available years (period) | AAD [CHF m.] | Event damage with 5-year RP [CHF m.] | Event damage with 10-year RP [CHF m.] | Event damage with 50-year RP [CHF m.] | Event damage with 250-year RP [CHF m.] | Event damage due to Lothar/ Martin [CHF m.] |
|---|---|---|---|---|---|---|---|
| Insurance claims data | 34 (1981-2014) | 2.3 | 0.6 | 1.1 | - | - | 62.4 |
| "WISC historic" | 75 (1940-2014) | 1.4 | 0.2 | 1.3 | 31.4 | - | 62.7 |
| "WISC probabilistic extension" | 2'250 (30*75) | 1.4 | 0.2 | 1.3 | 17.0 | 74.6 | - |

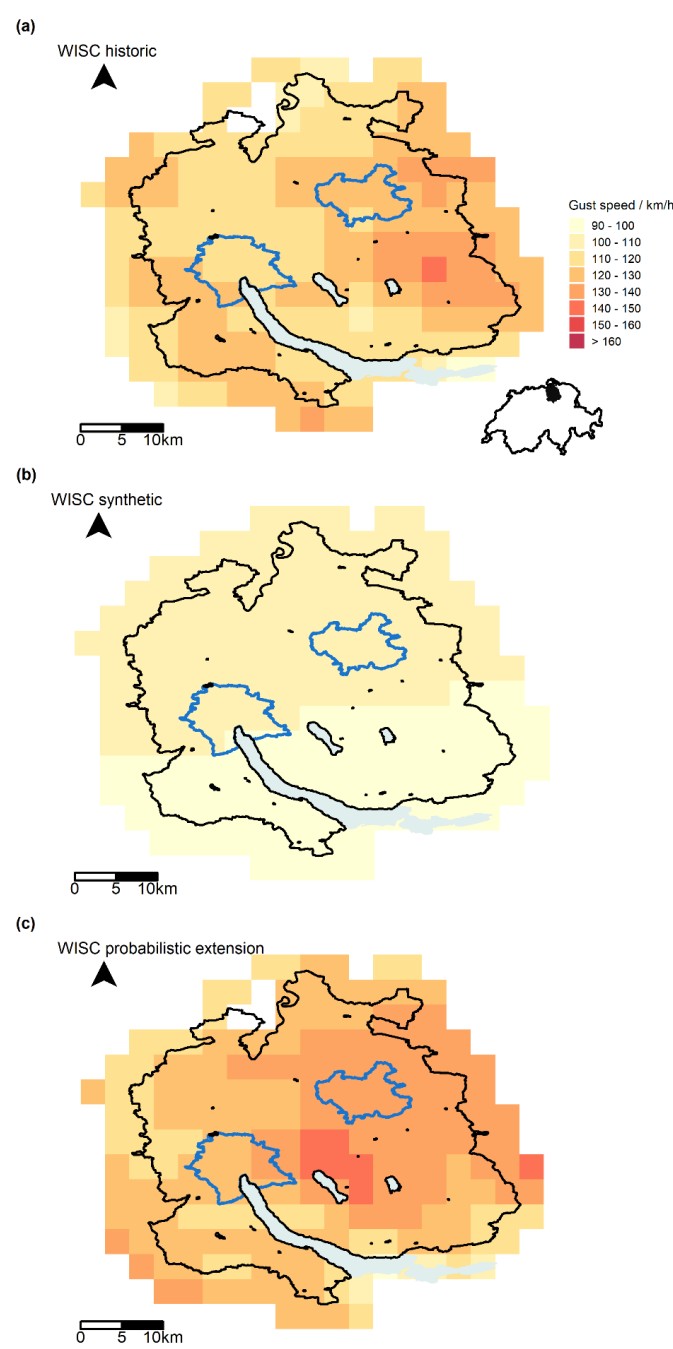

**Figure 1:** Maximum wind gusts for every grid cell in the canton of Zurich (i.e., windstorm footprints) for the most damaging events in **(a)** "WISC historic", **(b)** "WISC synthetic", and **(c)** "WISC probabilistic extension". The urban areas of the two main cities Zurich (left) and Winterthur (right) are marked in blue.

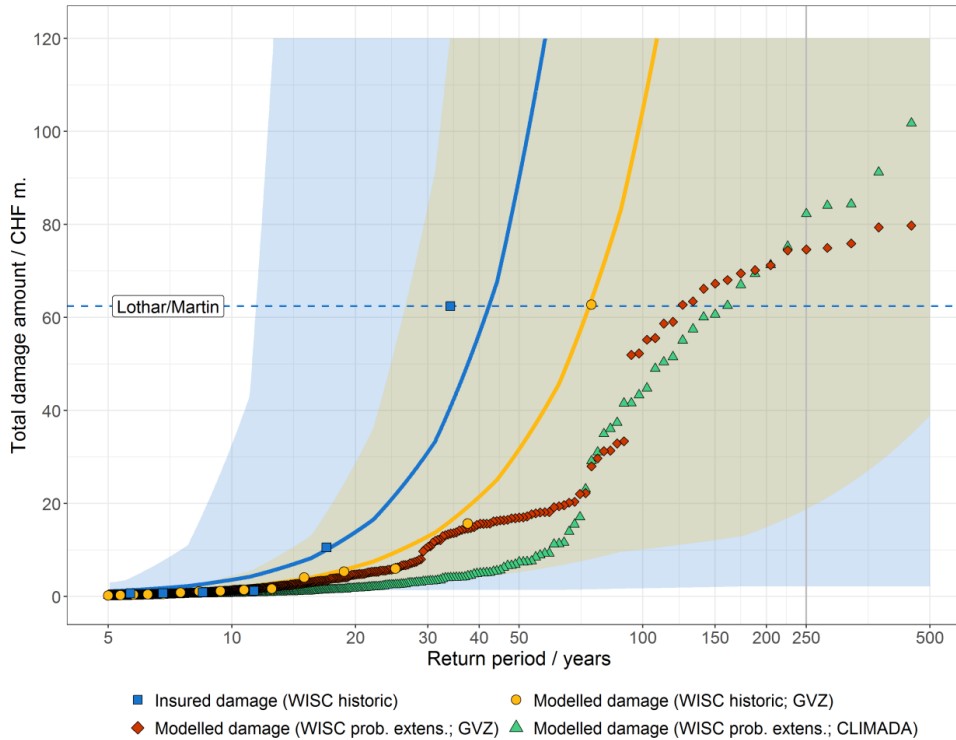

**Figure 2:** Exceedance frequency curves for building damages in the canton of Zurich based on different data sources. The blue squares indicate the insured damages according to GVZ's database (excluding the additional windstorms in 2017 and 2018), the blue solid line represents a GPD fitted to the insured damages, and the blue ribbon is the 90-% confidence interval produced by resampling. The yellow dots, solid line, and ribbon are analogous to the blue, but for the modelled damages based on "WISC historic" and the GVZ damage model. The red diamonds (green triangles) show the exceedance frequency curve of the modelled damages based on the hazard event set "WISC probabilistic extension" and the GVZ damage model (CLIMADA). The insured total damage for Lothar/Martin is shown by a blue dashed horizontal line, and the 250-year return period is indicated by a grey solid vertical line.



**Figure 3:** 2d-histograms for the normalised insured total damages in the canton of Zurich versus the modelled total damages based on **(a)** the GVZ damage model (diamonds) and **(b)** the CLIMADA impact model (triangles), respectively, for all windstorms with damage > 0 in the hazard event set "WISC historic". Marginal histograms are shown in the top and right panels.



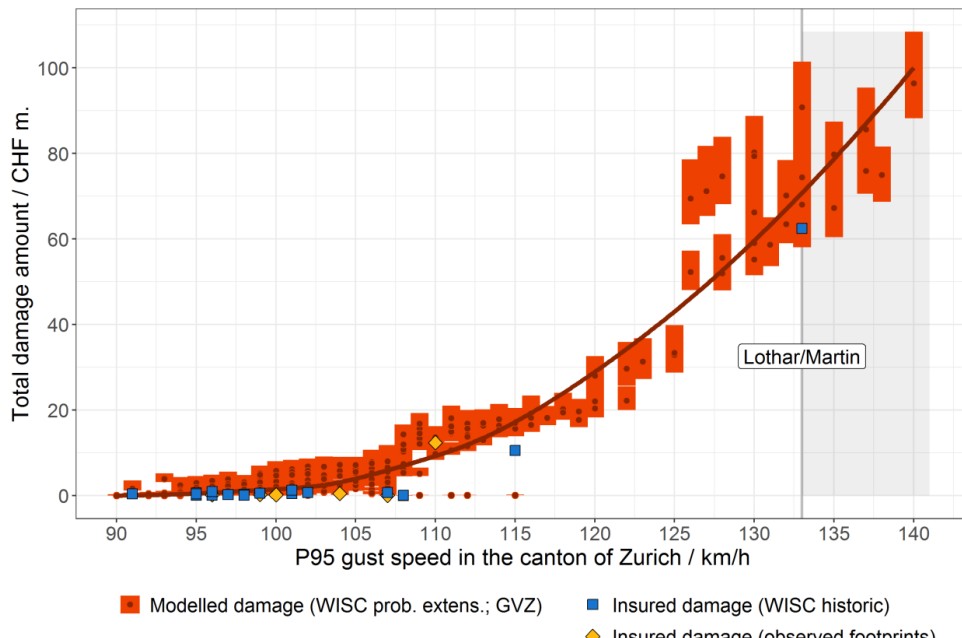

**Figure 4:** Total damage modelled using the GVZ damage model and the hazard event set "WISC probabilistic extension" versus the 95th percentile of the corresponding gust speeds in the canton of Zurich (median of 1'000 random damage modelling as red points; range of modelled damages indicated as red bars). The 95th percentile of the gust speeds is shown, because the 95th percentile is used in GVZ's damage model to categorise windstorm events (Sect. 2.3.1). The relationship between wind gust speed and modelled total damage is further approximated by a locally estimated scatterplot smoothing (LOESS) and a bootstrap method (i.e., random resampling with replacement, number of samples = 1'000; median of confidence interval given as solid red line). Furthermore, the relationship between gust speeds and normalised insured total damages based on "WISC historic" and independent, interpolated wind gust observations (selection of windstorms in 2017 and 2018, including winter windstorm Burglind) are given as blue squares and yellow diamonds, respectively. The domain for unprecedented windstorms − i.e. beyond Lothar/Martin − is shaded grey.





**Appendix**

**Table A1:** AAD and event damage for different return periods (RP) and the windstorm event Lothar/Martin on the basis of insurance claims data and modelled damages using the CLIMADA impact model and the hazard event sets "WISC historic" and "WISC probabilistic extension", respectively.

|  | Available years (period) | AAD [CHF m.] | Event damage with 5-year RP [CHF m.] | Event damage with 10-year RP [CHF m.] | Event damage with 50-year RP [CHF m.] | Event damage with 250-year RP [CHF m.] | Event damage due to Lothar/ Martin [CHF m.] |
|---|---|---|---|---|---|---|---|
| Insurance claims data | 34 (1981-2014) | 2.3 | 0.6 | 1.1 | - | - | 62.4 |
| "WISC historic" | 75 (1940-2014) | 1.1 | 0.2 | 0.6 | 24.5 | - | 62.6 |
| "WISC probabilistic extension" | 2'250 (30*75) | 1.2 | 0.2 | 0.6 | 7.4 | 82.3 | - |

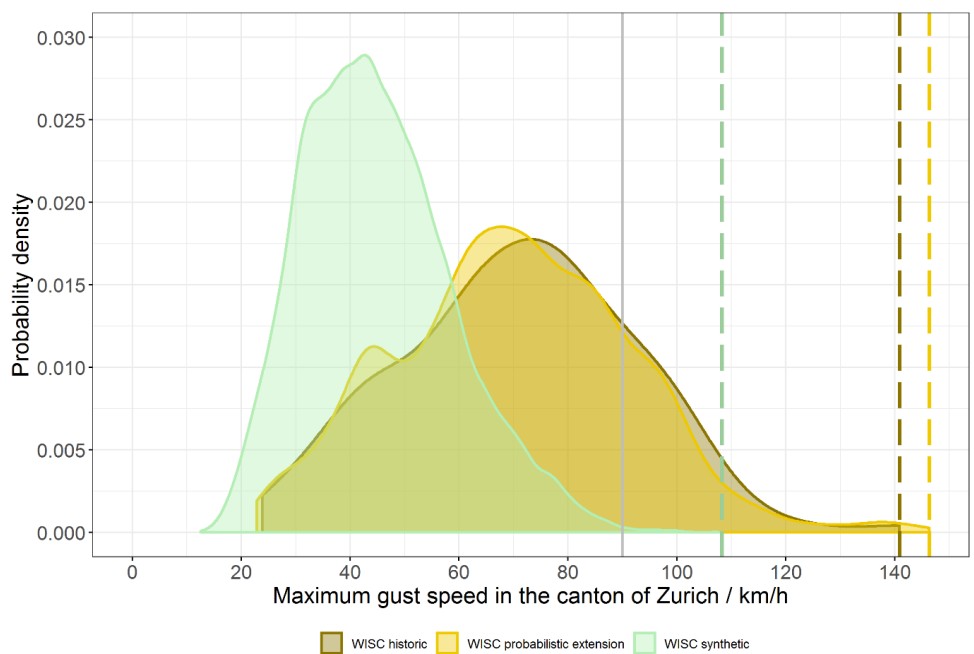

792

**Figure A1:** Probability density functions of the maximum gust speeds at building level in the canton of Zurich
for the three hazard event sets "WISC historic" (brown), "WISC probabilistic extension" excluding the parent
windstorms (yellow), and "WISC synthetic" (green). The maxima of the individual distributions are shown as
dashed vertical lines. In the GVZ damage model, damage is possible from a wind gust speed of more than
90 km/h, which is here indicated by a grey solid vertical line.


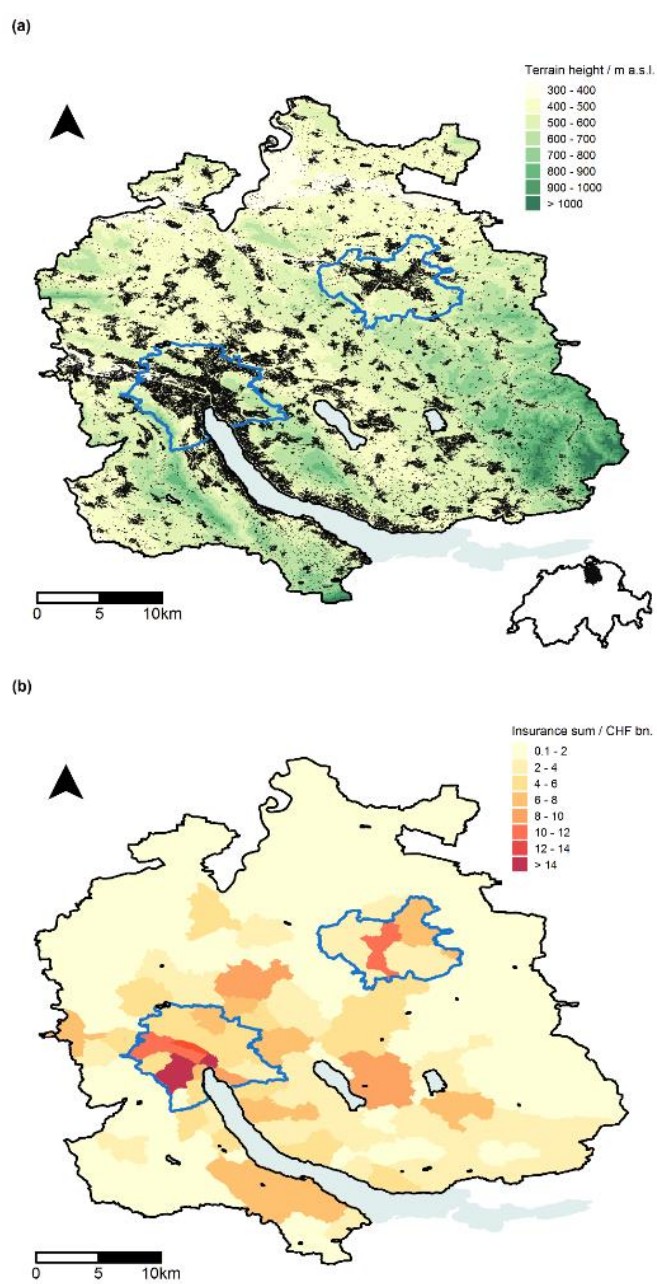

**Figure A2: (a)** Terrain height for the canton of Zurich (colour scheme) according to a digital elevation model
with a horizontal grid size of 200 m (Source: Swiss Federal Office of Topography; Swisstopo DEM, 2019). In
addition, the spatial distribution of all buildings insured by GVZ is indicated and the urban areas of the two
main cities, Zurich (left) and Winterthur (right), are marked in blue. **(b)** Total building sum insured for each
municipality (colour scheme).

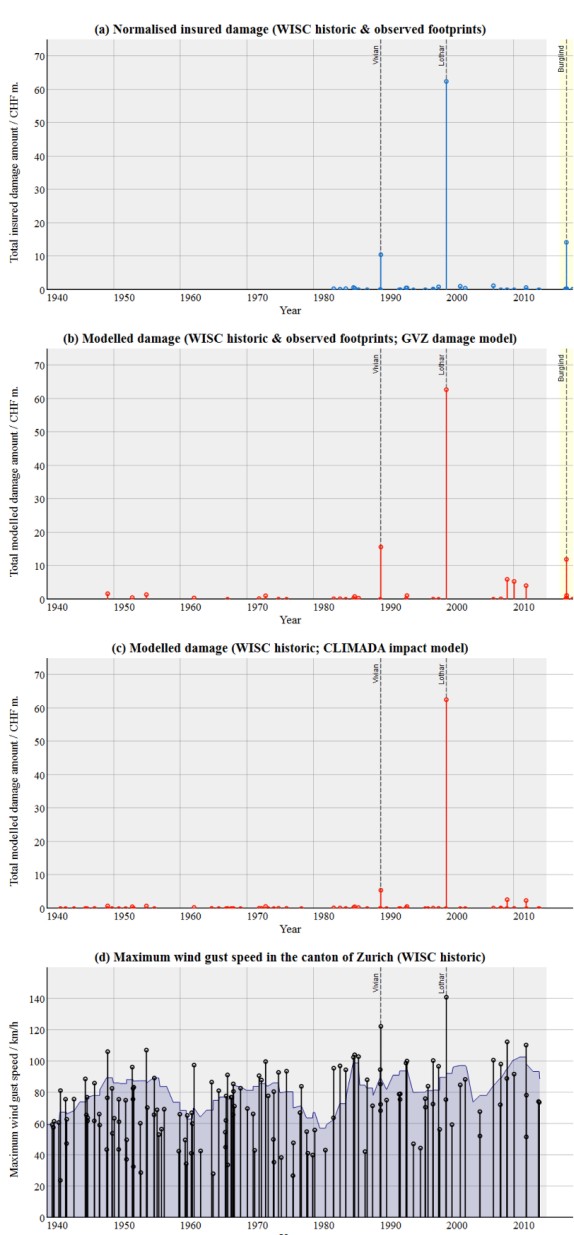

804

**Figure A3:** Variability of windstorms and associated damages in the canton of Zurich: **(a)** normalised insured
damage, **(b)** modelled windstorm damage based on the GVZ damage model and the hazard event sets "WISC
historic" and "observed footprints", **(c)** modelled windstorm damage based on the CLIMADA impact model and
"WISC historic", and **(d)** maximum gust speeds at building level in the canton of Zurich according to "WISC
historic" (black stem plot). The filled time series in (d) additionally shows the 5-year moving average of the
yearly maximum gust speeds in the canton of Zurich. The period for which "WISC historic" hazard data
("observed footprints") is available is shaded grey (yellow) in (a) and (b). The windstorm events
Vivian/Wiebke, Lothar/Martin, and Burglind are marked.