# Peer review of "Comparing an insurer's perspective on building damages with"

_Natural Hazards and Earth System Sciences, 2020_

## Referee Comment (RC1) · Alexandros Georgiadis (Referee) · 25 May 2020

The main objective of the paper is to demonstrate the value of catastrophe modelling analysis in respect of estimating the frequency of high intensity storms, compared to a pure statistical analysis of the claims history from a portfolio that has a limited record of a few decades. Two catastrophe models with different vulnerabilities and exposures are used to calculate the losses, GVZ's proprietary model and the open source CLIMADA platform. Both models perform very well in calculating the losses of a numbers of historical storms (e.g. Vivian, Lothar, Burglind) so are clearly appropriate tools

for the stated job. The selected hazard inputs include the: (i) WISC historical set of 75 years with 142 events and (ii) a probabilistic perturbation of the above event set, where every storm has 29 altered offsprings, thus the set is extended to have 4,260 storms covering 2,250 years. On the other hand, the insured claims dataset consists of about 40 years of losses that provides 18 storms which are available in the WISC historical set. Overall, I think that the presented work is of high quality: there are no obvious methodological errors and the findings are robust regarding the stated purpose, to complement claims-based risk assessment with a modelling approach. The conclusion that the return period of intense storms (like Lothar) cannot be determined sufficiently from a simple analysis of claims history is robust, but also well established in the Insurance industry. The proposed approach to produce a probabilistic event set by perturbing/expanding the WISC historical events, then calculate the losses using one or more damage models is technically correct and appropriate but it is not novel. Focusing on the results, I think that risk assessment at the tail will benefit from an attempt to build a more focused estimation of the uncertainty associated with the WISC probabilistic exceedance probability curves in Figure 2. The confidence interval based on the WISC historical set (CHF 19M to 33000M) is very conservative and negates much of the fundamental advantage of complementing risk assessment with probabilistic catastrophe modelling. I think that this is the major point to be addressed in the analysis, thus I would recommend publishing the article conditionally the authors provide a substantial response to this question (see below, bullet points: 2.a-c). Also, suggestions to further expand the work (beyond the scope of the current article) are available in the end of bullet point 1.

More specifically, I will address the following scientific question/issues:

1. The proposed approach to produce a probabilistic event set by perturbing/expanding the WISC historical events is technically correct and appropriate given the scope of the analysis. Having said that, although acceptable, the approach is not novel. Several (re)insures have proprietary cat models that follow similar methodologies. A limited

historical 'seeding' data-set (often based on reanalysis data, e.g. 20C_R, ERA-Int, ECMWF_R) is extended either by a statistical perturbation/resampling approach (e.g. Swiss Re) or extensive use of dynamical modelling (usually regional climate modelling-RCM) outputs (e.g. Weather Predict/Renaissance Re, Partner Re) to produce a realistic probabilistic event set. The advantage of the latter is the physical consistency of each individual stochastic event due to the physics-based simulation of the RCM. Furthermore, the main catastrophe model vendors in the market (RMS, AIR, AON Impact forecasting and more) tend to provide probabilistic windstorm solutions based on outputs extracted for a variety of long global climate model (GCM) runs, calibrated (often fitted) against the available historical record. The advantage of this approach is that the simulation generates physically realistic storms that are not constrained by the attributes/parameters of the seeding historical windstorms. Such methodologies directly address the main limitation of the WISC probabilistic expansion approach used by the authors that results to almost identical AAD values in Tables 2 (1.4M CHF) and A1 (1.1M-1.2M CHF) for the WISC historic and probabilistic sets. The probabilistic expansion adds very little further risk hazard information compared to the seeding historical set. A possible avenue for the authors to continue the current work would be to look into calibrating the WISC synthetic gusts distribution (in figure A1, lines 793-797) against the WISC historical event set to address the low gust speed intensity. Then repeat the loss calculation with the 'enhanced' WISC synthetic event set.

2. The approach to expand the WISC historical events and determine the frequencies of the offspring probabilistic storms (GEV distribution fitted to the historical SSI values) has merit, and the concluding results in paragraphs 3.2 and 3.3, also provided in table 2, are realistic. I am not surprised the two WISC-based analyses reduce the calculated AAD value between 1.1 and 1.4M CHFs. Also, Lothar/Martin's return period is (correctly) positioned at and above 75 yrs, potentially beyond 125 yrs. Considering the disproportional yet uncertain impact of the extreme event Lothar/Martin on the claims data analysis, the above results are plausible, yet the authors do not follow with a narrower estimation of the uncertainties. I understand why the authors prefer to retain the confidence interval based on the WISC historical set (CHF 19M to 33,000M), yet this reduces somewhat the functionality of the probabilistic expansion model. It's main objective is to provide a tail view. Here are a few suggestions: (a) The 4,260 storms in the WISC probabilistic set provide the equivalent of 2,250 years of storm activity (based on the analysis assumptions). You may sample randomly the equivalent of 250 or 500 years of storms and build multiple exceedance frequency curves for each sample. A spaghetti plot of the 'secondary' exceedance frequency curves will enable a reviewed estimation of the uncertainty around the curve. Essentially the idea is not dissimilar to the re-sampling approach described in paragraph 2.4.3 for the Pareto Pricing. (b) Estimate multiple probabilistic extensions of the WISC historic event set with different initial assumptions including (but not limited to) fitting different extreme distributions (e.g. Weibull, Pareto), inclusion/exclusion of Lothar/Martin in the seeding WISC historic set to quantify the sensitivity of the methodology in the most extreme event in the set, for both damage models (GVZ & CLIMADA). This will produce an ensemble of exceedance frequency curves that can be visualized as a spaghetti plot. (c) A combination of the above two ideas can work as well.

3. One aspect which is underrepresented in the discussion is the role of the loss uncertainty due to the vulnerability (and exposure) components. GVZ's damage model has a stochastic component as seen in figure 4, also described in the text (lines 443 to 449), yet it is unclear whether the damage (given by the red bars in figure 4) informs the process of building the exceedance frequency curve of the modeled damage based on the WISC probabilistic extension of figure 2. Please clarify.

4. The two modelling approaches (GVZ damage model & CLIMADA impact model) use different input exposures as described in lines 272 for GVZ's model and 303 for CLIMADA. Is it possible to get a feeling regarding the difference between the two input exposures (e.g. 10%, 50%)?

[Figure]

2020-115, 2020.

---

## Referee Comment (RC2) · Anonymous Referee #2 · 18 Jun 2020

**General comments**

This paper compares windstorm risk estimations (such as annual average damage, exceedance frequency curves) in the canton of Zurich, Switzerland, using insurance claims data, and modelled damages with two models (GVZ and CLIMADA) using various hazard inputs ('WISC historic' and 'WISC probabilistic extension'). They find that the claims data is skewed by the extreme event Martin/Lothar, leading to a shorter return period for that storm and higher average annual damages compared to the results from the longer modelled datasets.

The paper is well written and the results are worthy of publication. My main issue is that I feel the conclusions about return periods derived from 'WISC probabilistic' may have been overstated. The authors correctly state in their discussion (L486-499), the 'WISC probabilistic' dataset does not reduce uncertainty compared to 'WISC historic' because they're based on the same data, but in some instances I think it is important to emphasise the uncertainty (I include examples in the 'specific comments' below).

**Specific comments**

1. Abstract L20: "Additionally, the probabilistic modelling approach allows assessing rare events, such as a 250-year return period windstorm causing CHF 75 million damages" – please emphasise the uncertainty here.

2. Section 2.2.2: I don't think it's necessary to describe 'WISC operational' and 'WISC stochastic' as they are not used. It is already mentioned in the introduction why you can't use 'WISC stochastic' (L102; perhaps you could refer to fig A1 here), and the reasons for not using 'WISC operational' could also be discussed here.

3. Section 2.2.3 L209: please could you mention here that you describe how alpha and beta are chosen later in the section?

4. Equations (1) (L209-210): I presume this transformation is applied at each grid point, so that a wind speed from a grid point $i$ becomes the $windspeed_{original}$ at grid point $j$ in the shifted footprint? If so, how do you account for different properties of grid points $i$ and $j$ – for example, they could have very different roughness and altitudes (in an extreme case $i$ could be over open water and $j$ could be in a sheltered area, so would have much lower expected wind speeds).

5. L215/216: The references given for the storm severity index all have different definitions. Which formula did you use here?

6. L282-287: This paragraph is a bit confusing. I guess you mean to say that MDD is calculated from the vulnerability curve of Schwierz et al, and you use this same vulnerability curve in CLIMADA?

7. L348/349: How many data points did you have above the threshold in each case? When you do the re-sampling, is the number of re-sampled points (200) equal to the number of points you used for the original fit?

8. Section 3.3: L386-391: I think you need more emphasis on the uncertainty in the return period of Lothar/Martin. Although the value from the claims is much smaller (34yrs), it's still within the 90% confidence interval from WISC historic (25yrs to > 500 yrs)

9. L398: Again, I think you should mention that the 250yr RP from the claims data is within the range estimated from WISC historic.

10. L400-404: Since the 'WISC probabilistic extension' and 'WISC historic' 250yr RPs are well within the 90% confidence intervals of one another, can you really conclude anything about the difference in return periods?

11. Section 3.5 L439-440: "In total, "WISC probabilistic extension" contains 17 events which are potentially more damaging than Lothar/Martin": I assume the 17 events referred to in the text are the 17 red dots in Fig 4 with damages > Martin/Lothar damage, rather than the events with P95 gusts speed > P95 gust speed of Martin/Lothar, so shouldn't the grey area in Fig 4 be bounded by a horizontal line at damage ≈ CHF 62m, rather than the vertical line at P95 gust speed ≈ 133km/h?

12. L441: "A (modelled) total damage amount of more than CHF 96 million is associated with the most extreme windstorm event in "WISC probabilistic extension"": In Fig 2 it looks like the highest damage storm in "WISC probabilistic extension" has a damage amount of approximately CHF 80m. Why is the maximum damage in Fig 4 higher? Aren't they the same storms?

13. Fig 4: Please could you clarify if the insured damages (blue squares and yellow diamonds) are the values from the claims dataset (after normalising), or the damage amounts estimated from the GVZ model on the historical events?

14. Fig 4: Please could you explain why there are quite a few footprints from WISC probabilistic with zero damage despite having P95 gust speeds of 107-115 km/h? Is it because they mainly hit unpopulated areas?

---

## Author Comment (AC1) · 28 Jul 2020

**AUTHORS' RESPONSE TO REFEREE #1**

**Research article:**

Comparing an insurer's perspective on building damages with modelled damages from pan-European winter windstorm event sets: a case study from Zurich, Switzerland (Nat. Hazards Earth Syst. Sci. Discuss., https://doi.org/10.5194/nhess-2020-115, in review; submitted on 07 April 2020)

**Authors:**

Christoph Welker, Thomas Röösli, David N. Bresch

*We thank the referee Dr. Alexandros Georgiadis for his comments, which have improved the quality of the manuscript.*

*The original comments from the referee are listed below directly followed by our responses in blue and italic and changes to the manuscript in blue and bold.*
* * *
The main objective of the paper is to demonstrate the value of catastrophe modelling analysis in respect of estimating the frequency of high intensity storms, compared to a pure statistical analysis of the claims history from a portfolio that has a limited record of a few decades. Two catastrophe models with different vulnerabilities and exposures are used to calculate the losses, GVZ's proprietary model and the open source CLIMADA platform. Both models perform very well in calculating the losses of a numbers of historical storms (e.g. Vivian, Lothar, Burglind) so are clearly appropriate tools for the stated job. The selected hazard inputs include the: (i) WISC historical set of 75 years with 142 events and (ii) a probabilistic perturbation of the above event set, where every storm has 29 altered offsprings, thus the set is extended to have 4,260 storms covering 2,250 years. On the other hand, the insured claims dataset consists of about 40 years of losses that provides 18 storms which are available in the WISC historical set. Overall, I think that the presented work is of high quality: there are no obvious methodological errors and the findings are robust regarding the stated purpose, to complement claims-based risk assessment with a modelling approach. The conclusion that the return period of intense storms (like Lothar) cannot be determined sufficiently from a simple analysis of claims history is robust, but also well established in the Insurance industry. The proposed approach to produce a probabilistic event set by perturbing/expanding the WISC historical events, then calculate the losses using one or more damage models is technically correct and appropriate but it is not novel. Focusing on the results,

I think that risk assessment at the tail will benefit from an attempt to build a more focused estimation of the uncertainty associated with the WISC probabilistic exceedance probability curves in Figure 2. The confidence interval based on the WISC historical set (CHF 19M to

33000M) is very conservative and negates much of the fundamental advantage of complementing risk assessment with probabilistic catastrophe modelling. I think that this is the major point to be addressed in the analysis, thus I would recommend publishing the article conditionally the authors provide a substantial response to this question (see below, bullet points:

2.a-c).

*In the eyes of both referees, the uncertainty of the probabilistic event set "WISC probabilistic*

*extension" should be discussed in more detail. Nonetheless, they have different opinions about*

*the uncertainty estimations: While Referee #1 writes that "the confidence interval based on the*

*WISC historical set [...] is very conservative and negates much of the fundamental advantage of*

*complementing risk assessment with probabilistic catastrophe modeling", Referee #2 writes that*

*"the authors correctly state in their discussion [...], the 'WISC probabilistic' dataset does not*

*reduce uncertainty compared to 'WISC historic' because they're based on the same data".*

*The opposing ways of interpretation of both referees show that there are obviously different ways*

*of interpretation about whether the uncertainty of risk assessment can be reduced by a*

*probabilistic event set based on the same data. In a way, we represent the "conservative" way of*

*interpretation in our paper, i.e. that the uncertainty cannot be reduced by statistical*

*perturbation, and we would like to continue to support this way of interpretation. We will discuss*

*this further below.*

*In this response, we will show that the illustration of uncertainty, as requested by Referee #1,*

*partly ignores the parameter uncertainty and that is why the full uncertainty cannot be*

*illustrated easily. We will mention the uncertainty more often as requested by Referee #2.*

*In the following, we would like to briefly clarify our way of interpretation of the uncertainty*

*associated with historic and probabilistic event sets in general and in the case of this paper:*

*(1) Historic event sets:*

*Regarding the risk from rare events, an important source of uncertainty is the sampling*

*uncertainty. In this paper, we illustrate the sampling uncertainty of both insurance claims*

*data and modelled damages based on "WISC historic" by showing the 90-% confidence*

*interval derived by resampling (see Fig. 2).*

*(2) Probabilistic event sets:*

*As Referee #1 summarises, a probabilistic event set can be generated by statistical*

*perturbation and by using dynamical models. The sources of uncertainty are different for*

*both approaches. In the following, we only want to discuss statistical perturbation, as this*

*was the subject of the paper. We used statistical perturbation with two parameters with*

*the aim of representing the distribution of pan-European windstorm severity. By doing*

*this for the best-fit distribution, we transformed the sampling uncertainty of the severity*

*of historic windstorm events into parameter uncertainty of our model. However, as our*

*statistical approach does not add any additional information, the uncertainty is finally*

*not reduced. In our opinion, only if the process of generating a probabilistic event set*

*does include additional information one could argue in favour of the probabilistic event*

*set reducing uncertainty.*

*As this interpretation and argumentation needs to be clarified in the manuscript, we will*

*incorporate it at different points throughout the revised manuscript.*

Also suggestions to further expand the work (beyond the scope of the current article) are available in the end of bullet point 1.

More specifically, I will address the following scientific question/issues:

(1) The proposed approach to produce a probabilistic event set by perturbing/expanding the

WISC historical events is technically correct and appropriate given the scope of the analysis. Having said that, although acceptable, the approach is not novel. Several (re)insures have proprietary cat models that follow similar methodologies. A limited historical 'seeding' data-set (often based on reanalysis data, e.g. 20C_R, ERA-Int,

ECMWF_R) is extended either by a statistical perturbation/resampling approach (e.g.

Swiss Re) or extensive use of dynamical modelling (usually regional climate modelling-

RCM) outputs (e.g. Weather Predict/Renaissance Re, Partner Re) to produce a realistic probabilistic event set. The advantage of the latter is the physical consistency of each individual stochastic event due to the physics-based simulation of the RCM. Furthermore, the main catastrophe model vendors in the market (RMS, AIR, AON Impact forecasting and more) tend to provide probabilistic windstorm solutions based on outputs extracted for a variety of long global climate model (GCM) runs, calibrated (often fitted) against the available historical record. The advantage of this approach is that the simulation generates physically realistic storms that are not constrained by the attributes/parameters of the seeding historical windstorms.

*As the referee rightly states, there are many different ways to assess the risks from European winter windstorms. We show two possible approaches in this paper, i.e. a methodology implemented in a proprietary model and one in an open source model, and discuss which uncertainties have to be considered with these two approaches. Furthermore, we check the reliability of the open source impact model CLIMADA with both GVZ's claims data and output from their proprietary damage model. Those kinds of proprietary data are usually not available for scientific publications.*

*The paper was not necessarily about showing a new methodology. In our view, the recent development of freely accessible data on windstorm footprints (WISC) in combination with an open source damage model (CLIMADA) opens up new opportunities for applied research and provides a straightforward entry point for insurance companies to model the risks associated with winter windstorms in Europe – thus providing an additional / alternative perspective compared to inhouse or commercial models (as listed by the referee above). The application example we give is something new because of the open source concept presented.*

Such methodologies directly address the main limitation of the WISC probabilistic expansion approach used by the authors that results to almost identical AAD values in Tables 2 (1.4M CHF) and A1(1.1M-1.2M CHF) for the WISC historic and probabilistic sets. The probabilistic expansion adds very little further risk hazard information compared to the seeding historical set. A possible avenue for the authors to continue the current work would be to look into calibrating the WISC synthetic gusts distribution (in figure A1, lines 793-797) against the WISC historical event set to address the low gust speed intensity. Then repeat the loss calculation with the 'enhanced' WISC synthetic event set.

*We thank the referee for his suggestion to calibrate the distribution of the event set*

*"WISC synthetic" against "WISC historic". However, we do not think that this would be*

*successful in the case of "WISC synthetic" for the following reason.*

*The event set "WISC synthetic" contains wind gust footprints for around 23'000 synthetic*

*windstorms: i.e., three sets of 7'660 events each. In each of the three sets a different*

*approach was applied to carry out a calibration (see*

*[https://wisc.climate.copernicus.eu/wisc/#/help/products#eventset_section](https://wisc.climate.copernicus.eu/wisc/#/help/products#eventset_section)), which*

*however ultimately did not solve the problem of a generally lower severity of the*

*synthetic windstorm events compared to the historic ones. It is possible, that such a*

*calibration would be more successful, if applied to hourly wind gust data, before the*

*aggregation to 72-hour events is done. This is analogous to the conclusion about*

*correcting WISC wind gust data for higher altitudes in Marseille et al. (2017).*

*We agree with the referee, that in general, a probabilistic event set originating from*

*dynamical modelling could provide new information and would allow to reduce the*

*uncertainty, which is the main limitation of our WISC probabilistic expansion approach.*

*We think this is an important statement, that we would like to include as an outlook in the*

*revised manuscript. We suggest to add the following sentence at L495:*

"**In future studies, the information from dynamical models, which are run for many**

**model years, would help to further reduce this uncertainty.**"

(2) The approach to expand the WISC historical events and determine the frequencies of the offspring probabilistic storms (GEV distribution fitted to the historical SSI values) has merit, and the concluding results in paragraphs 3.2 and 3.3, also provided in table 2, are realistic. I am not surprised the two WISC-based analyses reduce the calculated AAD

value between 1.1 and 1.4M CHFs. Also, Lothar/Martin's return period is (correctly)

positioned at and above 75 yrs, potentially beyond 125 yrs. Considering the disproportional yet uncertain impact of the extreme event Lothar/Martin on the claims data analysis, the above results are plausible, yet the authors do not follow with a narrower estimation of the uncertainties. I understand why the authors prefer to retain the confidence interval based on the WISC historical set (CHF 19M to 33,000M), yet this
reduces somewhat the functionality of the probabilistic expansion model. It's main
objective is to provide a tail view. Here are a few suggestions:

a. The 4,260 storms in the WISC probabilistic set provide the equivalent of 2,250
years of storm activity (based on the analysis assumptions). You may sample
randomly the equivalent of 250 or 500 years of storms and build multiple
exceedance frequency curves for each sample. A spaghetti plot of the 'secondary'
exceedance frequency curves will enable a reviewed estimation of the uncertainty
around the curve. Essentially the idea is not dissimilar to the re-sampling
approach described in paragraph 2.4.3 for the Pareto Pricing.

b. Estimate multiple probabilistic extensions of the WISC historic event set with
different initial assumptions including (but not limited to) fitting different extreme
distributions (e.g. Weibull, Pareto), inclusion/exclusion of Lothar/Martin in the
seeding WISC historic set to quantify the sensitivity of the methodology in the
most extreme event in the set, for both damage models (GVZ & CLIMADA).
This will produce an ensemble of exceedance frequency curves that can be
visualized as a spaghetti plot.

c. A combination of the above two ideas can work as well.

*We thank the referee for his suggestions. We have implemented all of them and discuss*
*the results in the following. As a conclusion, we would still argue, that the yellow ribbon*
*in Fig. 2 (i.e., the sampling uncertainty of the modelled damages based on "WISC*
*historic") is the best illustration of the uncertainty for "WISC probabilistic extension".*
*We will include this argumentation in the manuscript, alongside the arguments already*
*provided in this response.*

*Following the referee's suggestion 2a and based on our data sample of total damages*
*modelled based on the hazard event set "WISC probabilistic extension" and the GVZ*
*damage model (red diamonds in Fig. 2), we sampled randomly the equivalent of*
*500 years of windstorms and built an exceedance frequency curve for each sample*
*(number of samples = 1'000). Accordingly, the red shading in Fig. R1-1 shows the 90-%*
*confidence interval as a result of the random resampling.*

[Figure]

Legend:
- ■ Insured damage (WISC historic)
- ● Modelled damage (WISC historic; GVZ)
- ◆ Modelled damage (WISC prob. extens.; GVZ)
- ▲ Modelled damage (WISC prob. extens.; CLIMADA)

*Figure R1-1: Modified Fig. 2. New is the red shading, which shows the 90-% confidence interval for the modelled damages based on "WISC probabilistic extension" and the GVZ damage model computed by applying the referee's suggestion 2a.*

*We are aware that the parameter uncertainty regarding the event set "WISC probabilistic extension" is important, especially in comparison with "WISC historic". However, in our opinion this source of uncertainty is not fully estimated and sufficiently illustrated with such a resampling methodology.*

*Following the referee's suggestion 2b, to include / exclude Lothar/Martin in the seeding, we tried a more systematic approach. We resampled (choice with replacement) the historic events (same number of events in each sample; choosing with replacement means some events are missing, whilst others are double). Then we created a probabilistic event set for each of these samples. The 90-% confidence interval is again given by the 5th and 95th percentiles of all samples. This is the best possible way we achieved to illustrate at least part of the uncertainty that originates from the fact that the best-estimate of the*

*distribution of the pan-European Storm Severity Index is unknown and thusly the*

*parameters for the creation of the probabilistic sets can only be chosen with a certain*

*degree of uncertainty. The uncertainty estimation up until a 30-year return period follows*

*approximately the uncertainty estimation for "WISC historic"; at higher return periods*

*the uncertainty estimation is levelling off, probably due to the limited ability of our*

*probabilistic approach to create very different (e.g., much stronger) events from the*

*seeding historic set. Therefore, we argue that the shown difference between the yellow*

*ribbon and the red ribbon could be misleading.*

[Figure]

***Figure R1-2:*** *Analogous to Fig. R1-1 but here the red shading shows the 90-%*

*confidence interval for the modelled damages based on "WISC probabilistic extension"*

*and the GVZ damage model computed by applying the referee's suggestion 2b.*

*The results for the referee's suggestion 2c, which is a combination of his suggestions 2a*

*and 2b, are given in Fig. R1-3. Firstly, we resampled (number of samples = 100) the*

*historic events and then used these different historic samples to create an ensemble of*

*probabilistic damage event sets (as suggested in 2b). Secondly, for each new*

*probabilistic damage event set, we sampled (number of samples = 20) randomly the*

*equivalent of 500 years of windstorm events and built an exceedance frequency curve for*

*each sample (as suggested in 2a). From this set of resampled and bootstrapped damage*

*event sets (total number of samples = 2000), we then calculated the 90-% confidence*

*interval. Whereas this combination provides a smooth illustration of the resampling*

*uncertainty, it still suffers from the same problem as the illustration in Fig. R1-2.*

*Therefore, we would still argue that the yellow ribbon in Fig. 2 is the best illustration of*

*the uncertainty for "WISC probabilistic extension".*

[Figure]

***Figure R1-3:*** *Analogous to Fig. R1-1 but here the red shading shows the 90-%*

*confidence interval for the modelled damages based on "WISC probabilistic extension"*

*and the GVZ damage model computed by applying the referee's suggestion 2c.*

(3) One aspect which is underrepresented in the discussion is the role of the loss uncertainty due to the vulnerability (and exposure) components. GVZ's damage model has a stochastic component as seen in figure 4, also described in the text (lines 443 to 449), yet it is unclear whether the damage (given by the red bars in figure 4) informs the process of building the exceedance frequency curve of the modeled damage based on the WISC
probabilistic extension of figure 2. Please clarify.

*The range of the modelled damages through the stochastic component in GVZ's damage*
*model (represented by red bars in Fig. 4) is not directly included in the calculation of the*
*exceedance probabilities in Fig. 2. Rather, we use the median of the damage range*
*modelled for each event to calculate the exceedance probabilities.*

*Additionally, we suggest to include the uncertainty related to vulnerability and exposure*
*in the following sentences at L510:*

**"A disadvantage of the used vulnerability curve is that it does not implicitly provide**
**a quantification of the uncertainty as a probabilistic vulnerability curve would (e.g.,**
**Heneka et al., 2006; Prahl et al., 2012). The quantification of the uncertainty of**
**exposure and vulnerability information was generally omitted in this study to focus**
**on the comparison of the claims and hazard datasets. But of course, for comparison**
**of the presented risk numbers with other studies the uncertainty of the vulnerability**
**and exposure information play a bigger role."**

The two references used have also been included at L659 and L688:

**"Heneka, P., Hofherr, T., Ruck, B., and Kottmeier, C.: Winter storm risk of**
**residential structures – model development and application to the German state**
**of Baden-Württemberg, Nat. Hazards Earth Syst. Sci., 6, 721–733,**
**doi:10.5194/nhess-6-721-2006, 2006."**

**"Prahl, B. F., Rybski, D., Kropp, J. P., Burghoff, O., and Held, H.: Applying**
**stochastic small-scale damage functions to German winter storms, Geophys.**
**Res. Lett., 39, L06806, doi:10.1029/2012GL050961, 2012."**

(4) The two modelling approaches (GVZ damage model & CLIMADA impact model) use
different input exposures as described in lines 272 for GVZ's model and 303 for
CLIMADA. Is it possible to get a feeling regarding the difference between the two input
exposures (e.g. 10%, 50%)?

  *The GVZ damage model uses an exposure information (i.e., insured value of the buildings*
  *in the canton of Zurich) which sums up to approximately 480 billion CHF. The exposure*
  *used in the CLIMADA impact model sums up to 80 % of that value.*

  *In this context, it is important to emphasise that differences in the total exposure values,*
  *compared to the GVZ damage model, were partially compensated by calibrating the*
  *damage functions in the CLIMADA impact model, in order be able to reproduce event*
  *damages comparable to those from the insurance claims database. We used publicly*
  *available exposure information in CLIMADA and not GVZ's proprietary portfolio*
  *information because of the open source concept presented in this paper. This way the*
  *presented methodology can be easily applied to other regions.*

  _________________________________________________________________________

**References used in this response**

*Heneka, P., Hofherr, T., Ruck, B., and Kottmeier, C.: Winter storm risk of residential structures*
  *– model development and application to the German state of Baden-Württemberg, Nat.*
  *Hazards Earth Syst. Sci., 6, 721–733, doi:10.5194/nhess-6-721-2006, 2006.*

*Marseille, G. J., Stoffelen, A., van den Brink, H., and Stepek, A.: WISC Bias Derivation and*
  *Uncertainty Assessment,*
  *https://wisc.climate.copernicus.eu/wisc/documents/shared/(C3S_441_Lot3_WISC_SC2-*
  *D3.3-CGI-RP-17-0071)%20(Final%20Bias%20Derivation)%20(v1.0).pdf, 2017.*

*Prahl, B. F., Rybski, D., Kropp, J. P., Burghoff, O., and Held, H.: Applying stochastic small-*
  *scale damage functions to German winter storms, Geophys. Res. Lett., 39, L06806,*
  *doi:10.1029/2012GL050961, 2012.*

---

## Author Comment (AC2) · 28 Jul 2020

**AUTHORS' RESPONSE TO REFEREE #2**

**Research article:**

Comparing an insurer's perspective on building damages with modelled damages from pan-
European winter windstorm event sets: a case study from Zurich, Switzerland (Nat. Hazards
Earth Syst. Sci. Discuss., https://doi.org/10.5194/nhess-2020-115, in review; submitted on
07 April 2020)

**Authors:**

Christoph Welker, Thomas Röösli, David N. Bresch

*We thank the referee for comments, which have improved the quality of the manuscript.*

*The original comments from the referee are listed below directly followed by our responses in*
*blue and italic and changes to the manuscript in blue and bold.*

___________________________________________________________________________

**General comments**

This paper compares windstorm risk estimations (such as annual average damage, exceedance
frequency curves) in the canton of Zurich, Switzerland, using insurance claims data, and
modelled damages with two models (GVZ and CLIMADA) using various hazard inputs ('WISC
historic' and 'WISC probabilistic extension'). They find that the claims data is skewed by the
extreme event Martin/Lothar, leading to a shorter return period for that storm and higher average
annual damages compared to the results from the longer modelled datasets.

The paper is well written and the results are worthy of publication. My main issue is that I feel
the conclusions about return periods derived from 'WISC probabilistic' may have been
overstated. The authors correctly state in their discussion (L486-499), the 'WISC probabilistic'
dataset does not reduce uncertainty compared to 'WISC historic' because they're based on the
same data, but in some instances I think it is important to emphasise the uncertainty (I include
examples in the 'specific comments' below).

*Referee #2 raised as main issue the uncertainty of the results derived from the hazard event set "WISC probabilistic extension". This is the same issue as raised by Referee #1. We would like to pick up the suggestions by Referee #2 and emphasise the uncertainty of our estimation more. We have also expanded our interpretation of the uncertainty in more detail in the response to Referee #1 and aim to clarify our interpretation of the uncertainty more clearly at different points throughout the manuscript.*

**Specific comments**

1. Abstract L20: "Additionally, the probabilistic modelling approach allows assessing rare events, such as a 250-year return period windstorm causing CHF 75 million damages" – please emphasise the uncertainty here.

   *We now emphasise the uncertainty in L20:*

   "Additionally, the probabilistic modelling approach allows assessing rare events, such as a 250-year return period windstorm causing CHF 75 million damages**, including an evaluation of the uncertainties**."

   *Please consider the given word limit for the abstract of a maximum of 200 words.*

2. Section 2.2.2: I don't think it's necessary to describe 'WISC operational' and 'WISC stochastic' as they are not used. It is already mentioned in the introduction why you can't use 'WISC stochastic' (L102; perhaps you could refer to fig A1 here), and the reasons for not using 'WISC operational' could also be discussed here.

   *To overcome the shortcomings of the event set "WISC synthetic", we propose in addition the probabilistic windstorm hazard event set "WISC probabilistic extension". We briefly describe "WISC synthetic", since we used "WISC synthetic" for a comparison with "WISC probabilistic extension" in Fig. 1 and Fig. A1. We also think that readers are asking themselves about calibrating the wind gust information from "WISC synthetic" as suggested by Referee #1. We would therefore like to provide this information in a structured way, for which the data and methods section is suitable for.*

   *"WISC operational" is described, basically to explain why we didn't use this event set in the context of the paper. We think that readers might ask themselves this question.*

*We would therefore like to keep this structure. Is that understandable?*

3. Section 2.2.3 L209: please could you mention here that you describe how alpha and beta are chosen later in the section?

*Yes, thank you for the hint. We suggest the following change to L207-209:*

*"The wind gust speeds were intensified and weakened by no more than 3 m/s (normally much less) according to the probabilistic alteration of wind speeds in Eq. (1), with a scale parameter α=0.0225 and a power parameter β=1.15* **(choice explained further below)***:* […]."

4. Equations (1) (L209-210): I presume this transformation is applied at each grid point, so that a wind speed from a grid point i becomes the windspeed$_{original}$ at grid point j in the shifted footprint? If so, how do you account for different properties of grid points i and j – for example, they could have very different roughness and altitudes (in an extreme case i could be over open water and j could be in a sheltered area, so would have much lower expected wind speeds).

*WISC wind gust footprints are available at a spatial resolution of 4.4 km. Small-scale changes in both topography and ground cover can indeed strongly influence the characteristics of wind gusts. However, those small-scale changes cannot be resolved sufficiently well in a model with a horizontal resolution of approximately 4 km. In general, the canton of Zurich is characterised by a gentle to moderate topography (see Fig. A2a). For these reasons, we decided not to make a correction regarding the topography (and the ground cover) in our current model setup.*

*Nonetheless, we think the referee touches on an important point and a refinement of our methodology would be interesting for a follow-up study. It is conceivable that the quality of the windstorm footprints from "WISC probabilistic extension" could be improved by using a correction method, which takes account of at least the topography.*

*In general, we think that the referee's comment is most relevant for those countries and regions in Europe which are characterised by a complex and pronounced topography or which border large water surfaces with a lower roughness compared to the land surface.*

*In the computation of the event set "WISC probabilistic extension", the spatial displacement was undertaken by shifting the respective windstorm footprint by about 20 km to the north, south, west, or east. For different regions and countries in Europe, we determined the difference in wind gusts, which results from this spatial displacement of the windstorm footprints. In total, there are 3'408 windstorm events that result from the spatial displacement of either the original windstorm footprints or of altered windstorm footprints (according to Eq. (1)).*

*Figure R2-1 shows for the canton of Zurich, the whole country of Switzerland, and the UK boxplots of all changes due to spatial displacement that occurred on any point in any event. Here, Switzerland and the UK were chosen as examples because Switzerland is a country which is characterised by a pronounced topography with high mountains and the UK is characterised by pronounced land-sea contrasts.*

[Figure]

*Figure R2-1:* *Boxplots of all differences due to spatial displacement that occurred on any point in any event for the canton of Zurich, the whole country of Switzerland, and the UK.*

*Figure R2-1 shows that the spatial displacements in windstorm footprints made can result in quite extreme changes in wind gust speed as one can see from changes of up to plus 16 m/s in the case of the canton of Zurich. These extremes are however very rare: 50 % of all points in all events are not changed by more than 2 m/s, and 90 % of all points are changed by no more than 5 m/s. The extremes are even higher in the case of Switzerland with up to 40 m/s and 25 m/s in the case of the UK. However, 50 % of all points in all events are not changed by more than 2 m/s in the case of Switzerland and*

*1 m/s in the case of the UK; 90 % of all points are changed by no more than 6 m/s in the case of Switzerland and 4 m/s in the case of the UK.*

*To be more precise in the paper, we have added this sentence to L211:*

"**In countries close to the sea or with a pronounced and high topography, the methodology for creating the probabilistic events might need adaptation to better incorporate the difference in surface roughness and altitude**."

5. L215/216: The references given for the storm severity index all have different definitions. Which formula did you use here?

*We used the formula described by Dawkins et al. (2016). We now emphasise this more strongly in L214-216:*

"In an effort to assign reasonable frequency estimates to the probabilistic windstorm footprints, we considered the distribution of the historic, pan-European Storm Severity Index (SSI; **formula used by Dawkins et al., 2016; further information in Lamb and Frydendahl, 1991; Leckebusch et al., 2008**)."

6. L282-287: This paragraph is a bit confusing. I guess you mean to say that MDD is calculated from the vulnerability curve of Schwierz et al, and you use this same vulnerability curve in CLIMADA?

*We have clarified the language in L282-287:*

"To estimate the damage in monetary terms, the value of each individual building (i.e., its insured value) was multiplied by the factor "Mean Damage Degree" (MDD**, a number between 0 and 1**) **calculated from the vulnerability curve of** Schwierz et al. (2010)**;** where the gust speeds at building level computed in the first step were converted into the corresponding MDD factors. The MDD factors are a non-linear function of the maximum wind gust speed during a windstorm event **and are diagrammed in Welker et al. (2016)**. The same vulnerability curve of Schwierz et al. (2010) is also implemented in the open source impact model CLIMADA (Aznar-Siguan and Bresch, 2019a)."

7. L348/349: How many data points did you have above the threshold in each case? When you do the re-sampling, is the number of re-sampled points (200) equal to the number of points you used for the original fit?

*The threshold defines how data points are used for the original fit. In the case of the insured damages, the threshold of CHF 0.4 million resulted in 9 data points above the threshold. In the case of the modelled damage event set based on "WISC historic", the threshold of CHF 0.1 million resulted in 19 data points above the threshold. As expressed in L346-347, these thresholds result "in a parameterised GPD with similar exceedance frequencies for the largest damage amount in the event set". Additionally, the number of data points per observation year is reasonably similar between these two damage event sets. The number of resampled points is equal to the number of data points we used for the original fit.*

*As mentioned in the "Code availability and data availability" section (L575-576), the code used for this analysis is published here:* [https://github.com/CLIMADA-project/climada_papers](https://github.com/CLIMADA-project/climada_papers).

8. Section 3.3: L386-391: I think you need more emphasis on the uncertainty in the return period of Lothar/Martin. Although the value from the claims is much smaller (34yrs), it's still within the 90% confidence interval from WISC historic (25yrs to > 500yrs)

*Thanks for the hint. We want to emphasise the uncertainty in our estimate of the return period of Lothar/Martin more and therefor suggest to insert the following sentences at the end of L391:*

**"These estimates represent the best guess for each damage event set. It is important to note that the quantified uncertainty of the estimate for the return period of Lothar/Martin based on "WISC historic" (yellow ribbon, 25 years to > 500 years) incorporates both the estimate for the insurance claims data (blue ribbon) as well as the estimate based on "WISC probabilistic extension"."**

*Additionally, we would add the following sentence to the discussion at L458:*

"We argue that the return period based on the historic windstorm footprints (75 years) is much more reliable than the return period based on the insured damage record (34 years).

**Well knowing that the two estimates each have overlapping uncertainties, the estimates do not contradict each other. Rather the estimates, as best guesses, can inform varying deterministic risk views.** Other information, like […]."

*In addition to this, we have made the following illustration, which we however do not show in the paper: Figure R2-2 shows in addition to the 90-% confidence interval the 50-% confidence interval, in order to show more clearly the change in the uncertainty range from the insurance claims data (blue ribbon) to the modelled damages based on "WISC historic" (yellow ribbon). As one would expect from the larger sample of windstorm events considered, the area of uncertainty is smaller in the case of the modelled damages based on "WISC historic" compared to the insured damages. Considering the 50-% confidence interval, the return period for the damage event Lothar/Martin is between approximately 25 and 250 years based on the claims data. For comparison, the estimate based on modelled damages using "WISC historic" provides a narrower uncertainty range between approximately 45 and 175 years. Based on the insurance claims data only, the return period for the damage event Lothar/Martin was estimated to be 34 years. Figure R2-2 shows that although this value is within the 90-% confidence interval it is not within the 50-% confidence interval from modelled damages using "WISC historic".*

[Figure]

*Figure R2-2: Modified Fig. 2. New is the darker blue and yellow inner shading, which shows the 50-% confidence interval for the insured damage and the damage modelled on the basis of "WISC historic".*

9. L398: Again, I think you should mention that the 250yr RP from the claims data is within the range estimated from WISC historic.

   *Thanks for the hint. We suggest to add the following sentence to the end of L399:*

   **"At a return period of 250 years, the quantified uncertainty of the estimate based on "WISC historic" incorporates both the estimate for the insurance claims data as well as the estimate based on "WISC probabilistic extension"."**

   Compare our answer to the referee's comment #8.

10. L400-404: Since the 'WISC probabilistic extension' and 'WISC historic' 250yr RPs are well within the 90% confidence intervals of one another, can you really conclude anything about the difference in return periods?

*We would like to clarify the language in L400-402:*

**"An interesting feature illustrated in Fig. 2 is that at higher return periods the**

**modelled damages on the basis of "WISC probabilistic extension" increase less**

**strongly compared to the two extrapolations based on the fitted distributions."**

11. Section 3.5 L439-440: "In total, "WISC probabilistic extension" contains 17 events which are potentially more damaging than Lothar/Martin": I assume the 17 events referred to in the text are the 17 red dots in Fig 4 with damages > Martin/Lothar damage, rather than the events with P95 gusts speed > P95 gust speed of Martin/Lothar, so shouldn't the grey area in Fig 4 be bounded by a horizontal line at damage ≈ CHF 62m, rather than the vertical line at P95 gust speed ≈ 133km/h?

*That's right, thanks for the hint. We have adjusted the figure accordingly (see also*

*Fig. R2-3). We agree that this adjustment better connects the text and the figure.*

[Figure]

***Figure R2-3:*** *Modified Fig. 4. New are the horizontal shading instead of the vertical one*

*and the label of the x-axis (see our response to the referee's comment #14).*

12. L441: "A (modelled) total damage amount of more than CHF 96 million is associated with the most extreme windstorm event in "WISC probabilistic extension": In Fig 2 it looks like the highest damage storm in "WISC probabilistic extension" has a damage amount of approximately CHF 80m. Why is the maximum damage in Fig 4 higher? Aren't they the same storms?

*For plotting reasons, the range of the y-axis in Fig. 2 was reduced in comparison to Fig. 4, since the area of uncertainty is very large in the case of large return periods > 500 years.*

13. Fig 4: Please could you clarify if the insured damages (blue squares and yellow diamonds) are the values from the claims dataset (after normalising), or the damage amounts estimated from the GVZ model on the historical events?

*In Fig. 4, the blue squares are the values from the claims dataset after normalising to present-day exposure levels for the period 1981-2014. The corresponding wind gust speeds on the x-axis are from the hazard event set "WISC historic". The yellow diamonds are also values from the claims dataset but for the period 2017-2018; the corresponding wind gust speeds on the x-axis are from the additional hazard event set "observed footprints for current windstorms" (see Sect. 2.2.4).*

14. Fig 4: Please could you explain why there are quite a few footprints from WISC probabilistic with zero damage despite having P95 gust speeds of 107-115 km/h? Is it because they mainly hit unpopulated areas?

*Yes, that's true. In GVZ's damage modelling approach, damage is possible from a wind gust speed of more than 90 km/h, and only buildings affected by such wind gusts were considered in the damage model. In the case of the four points that protrude in Fig. 4, the area with wind gust speeds > 90 km/h is only limited to a small region in the south of the canton of Zurich (see Fig. R2-4), with relatively few buildings potentially at risk. The modelled damage sums are not zero, but rather small (see Table R2-1).*

*In the case of the modelled windstorm footprints shown in Fig. R2-4, it is maybe not immediately obvious why the 95th percentile of the few buildings affected was calculated and shown in Fig. 4. The reason is as follows: When the GVZ damage model was developed and calibrated, this was done almost exclusively with observed windstorm*

*footprints that affected the entire canton of Zurich and, in principle, every building in the*

*canton was potentially affected – the "classic" so to speak, large-scale winter*

*windstorms. The 95th percentile of all potentially affected buildings turned out to be*

*suitable for this selection of windstorms to categorise them in a subsequent modelling*

*step. Based on this categorisation, a random sample of m buildings was selected*

*thereafter, with the number m depending on the windstorm's severity category and giving*

*a percentage of total affected buildings. Only those buildings with potential damage > 0*

*were considered in the following modelling steps. The model approach is therefore not*

*necessarily intended / calibrated for small-scale and modelled wind gust footprints.*

*To be more precise in Fig. 4, we have changed the labelling of the x-axis to:*

"P95 gust speed of **affected region** […]"

[Figure]

248 ***Figure R2-4:*** *Maximum wind gusts for every grid cell in the canton of Zurich for the*
249 *events with IDs 14113, 14112, 14114, and 14116 in the dataset "WISC probabilistic*
250 *extension". Wind gust speeds < 90 km/h are plotted in grey.*

***Table R2-1:*** *For the same events as in Fig. R2-4, total damage modelled using the GVZ*
*damage model (median of 1'000 random damage modelling) and the 95th percentile of*
*the corresponding gust speeds.*

| *Event ID* | *P95 wind gust speed / km/h* | *Median total damage amount / CHF m.* |
|---|---|---|
| *14113* | *109* | *0.07* |
| *14112* | *111* | *0.08* |
| *14114* | *112* | *0.08* |
| *14116* | *115* | *0.09* |

# Additional changes to the manuscript

*While editing the referee's comment #4, we noticed an error in Eq. (1) in the paper and we*
*would like to correct it in the revised manuscript. The correction ensures consistency between*
*the manuscript and the code used for the calculations. In the case of the definition of*
*windspeed$_{scenario\ 5}$, the sign of change was incorrectly reversed; two plus signs in the last line are*
*now corrected to two minus signs.*

*Eq. (1) is correctly defined as follows (L209-210):*

$$windspeed_{scenario\ 1} = windspeed_{original} + \alpha * windspeed_{original}{}^{\beta}$$

$$windspeed_{scenario\ 2} = windspeed_{original} - \alpha * windspeed_{original}{}^{\beta}$$

$$windspeed_{scenario\ 3} = windspeed_{original} + \alpha * \sqrt[\beta]{windspeed_{original}}$$

$$windspeed_{scenario\ 4} = windspeed_{original} - \alpha * \sqrt[\beta]{windspeed_{original}} \tag{1}$$

$$windspeed_{scenario\ 5}$$

$$= windspeed_{original} - \frac{\alpha}{2} * windspeed_{original}{}^{\beta} - \frac{\alpha}{2} * \sqrt[\beta]{windspeed_{original}}$$

---

## Referee Report (RR1)

Firstly, I want to thank the authors of the paper for their detailed and robust responses. I am very happy the authors payed a lot of attention to provide persuasive answers to all four research questions in my original review. Particular attention is given to respond to question (2), which was substantiated with an additional piece of analysis (regarding the uncertainty calculation) that I think will make a good addition to the paper.

Summaries of two (out of the four) main questions discussed in the review are listed below, along with the author's answers (in blue text), followed with my final comment/view in **bold text**.

 Review point: The proposed approach to produce a probabilistic event set by perturbing/expanding the WISC historical events is technically correct and appropriate given the scope of the analysis. Having said that, although acceptable, the approach is not novel. (...) the main catastrophe model vendors in the market (RMS, AIR, AON Impact forecasting and more) tend to provide probabilistic windstorm solutions based on outputs extracted from a variety of long global climate model (GCM) runs, calibrated (often fitted) against the available historical record. The advantage of this approach is that the simulation generates physically realistic storms that are not constrained by the attributes/parameters of the seeding historical windstorms.

Author's response: As the referee rightly states, there are many different ways to assess the risks from European winter windstorms. We show two possible approaches in this paper (...) The paper was not necessarily about showing a new methodology. In our view, the recent development of freely accessible data on windstorm footprints (WISC) in combination with an open source damage model (CLIMADA) opens up new opportunities for applied research and provides a straightforward entry point for insurance companies to model the risks associated with winter windstorms in Europe – thus providing an additional / alternative perspective compared to inhouse or commercial models (as listed by the referee above). The application example we give is something new because of the open source concept presented.

Reviewer's response: I understand and agree with the paper aim. You are not looking for a novel modelling methodology, instead, you provide an application example of how to extend the information available in the available WISC data and build an inhouse model. I think there is merit in your approach.

2. The approach to expand the WISC historical events and determine the frequencies of the offspring probabilistic storms (GEV distribution fitted to the historical SSI values) has merit, and the concluding results in paragraphs 3.2 and 3.3, also provided in table 2, are realistic. (...) I understand why the authors prefer to retain the confidence interval based on the WISC historical set (CHF 19M to 33,000M), yet this reduces somewhat the functionality of the probabilistic expansion model. It's main objective is to provide a tail view. Here are a few suggestions: (a) Sample randomly the equivalent of 250 or 500 years of storms and build multiple exceedance frequency curves for each sample. (b) Estimate multiple probabilistic extensions of the WISC historic event set with different initial assumptions ... (c) combination of the above two ideas.

Author's response: We thank the referee for his suggestions. We have implemented all of them and discuss the results in the following. As a conclusion, we would still argue, that the yellow ribbon in Fig. 2 (i.e., the sampling uncertainty of the modelled damages based on "WISC 168 historic") is the best illustration of the uncertainty for "WISC probabilistic

extension". We will include this argumentation in the manuscript, alongside the arguments already provided in this response. (...)

We are aware that the parameter uncertainty regarding the event set "WISC probabilistic extension" is important, especially in comparison with "WISC historic". However, in our opinion this source of uncertainty is not fully estimated and sufficiently illustrated with 184 such a resampling methodology. (...)

We resampled (choice with replacement) the historic events (...) Then we created a probabilistic event set for each of these samples. The 90-% confidence interval is again given by the 5th and 95th percentiles of all samples. This is the best possible way we achieved to illustrate at least part of the uncertainty that originates from the fact that the best-estimate of the distribution of the pan-European Storm Severity Index is unknown and thusly the parameters for the creation of the probabilistic sets can only be chosen with a certain degree of uncertainty. The uncertainty estimation up until a 30-year return period follows approximately the uncertainty estimation for "WISC historic"; at higher return periods the uncertainty estimation is levelling off, probably due to the limited ability of our probabilistic set. Therefore, we argue that the shown difference between the yellow ribbon and the red ribbon could be misleading. (...)

The results for the referee's suggestion 2c, which is a combination of his suggestions 2a and 2b, are given in Fig. R1-3. (...) Whereas this combination provides a smooth illustration of the resampling uncertainty, it still suffers from the same problem as the illustration in Fig. R1-2. Therefore, we would still argue that the yellow ribbon in Fig. 2 is the best illustration of the uncertainty for "WISC probabilistic extension".

Reviewer's response: Thanks for the extensive work that resulted in the uncertainty estimations given by the red ribbons in Fig R1-1 to 3. I think that this analysis illustrates very clearly three different levels of uncertainty estimations, blue, yellow and red ribbons. I should clarify that I do agree with your conclusion, the yellow ribbon gives the best account of the uncertainties associated with the "WISC probabilistic extension" and it should be included in the paper. This does not change my view though that the 'reduced' uncertainty in the red ribbon from the resampling approach also has merit and it should be included in the paper as well, not to replace the (yellow ribbon) full WISC probabilistic extension uncertainty, but to complement it. Yes, the uncertainty estimation from resampling is 'incomplete' yet it can be helpful in the practical case of model. The Uncertainty in the yellow ribbon is too broad to provide a comparison criterion between two different exceedance frequency curves from different models, (e.g. WISC hazard + GVZ versus WISC hazard + CLIMADA) Thus, I think inclusion of the resampling uncertainty (red ribbon) in addition to the full WISC probabilistic extension uncertainty ) yellow ribbon can be advantageous for your paper.

The remaining two discussion topics in the review (the role of the loss uncertainty due to the vulnerability and the different input exposures) has also been addressed thoroughly and I consider them clarified.

---

## Author Response (AR2)

**Research article:**

Comparing an insurer's perspective on building damages with modelled damages from pan-European winter windstorm event sets: a case study from Zurich, Switzerland (Nat. Hazards Earth Syst. Sci.

Discuss., https://doi.org/10.5194/nhess-2020-115, in review; submitted on 07 April 2020)

**Authors:**

Christoph Welker, Thomas Röösli, David N. Bresch

*We thank the referee Dr. Alexandros Georgiadis for his final comments, which have further improved the*

*quality of the paper.*

*The original comments from the referee are listed below directly followed by our responses in blue and*

*italic and changes to the manuscript in blue and bold.*
* * *
Firstly, I want to thank the authors of the paper for their detailed and robust responses. I am very happy the authors payed a lot of attention to provide persuasive answers to all four research questions in my original review. Particular attention is given to respond to question (2), which was substantiated with an additional piece of analysis (regarding the uncertainty calculation) that I think will make a good addition to the paper.

Summaries of two (out of the four) main questions discussed in the review are listed below, along with the author's answers […], followed with my final comment/view […].

Review point: The proposed approach to produce a probabilistic event set by perturbing/expanding the

WISC historical events is technically correct and appropriate given the scope of the analysis. Having said that, although acceptable, the approach is not novel. [...] the main catastrophe model vendors in the market (RMS, AIR, AON Impact forecasting and more) tend to provide probabilistic windstorm solutions based on outputs extracted from a variety of long global climate model (GCM) runs, calibrated (often fitted) against the available historical record. The advantage of this approach is that the simulation generates physically realistic storms that are not constrained by the attributes/parameters of the seeding historical windstorms.

Author's response: As the referee rightly states, there are many different ways to assess the risks from

European winter windstorms. We show two possible approaches in this paper [...] The paper was not necessarily about showing a new methodology. In our view, the recent development of freely accessible data on windstorm footprints (WISC) in combination with an open source damage model (CLIMADA)

opens up new opportunities for applied research and provides a straightforward entry point for insurance companies to model the risks associated with winter windstorms in Europe – thus providing an additional

/ alternative perspective compared to inhouse or commercial models (as listed by the referee above). The application example we give is something new because of the open source concept presented.

Reviewer's response: I understand and agree with the paper aim. You are not looking for a novel modelling methodology, instead, you provide an application example of how to extend the information available in the available WISC data and build an inhouse model. I think there is merit in your approach.

*Thanks for the positive comment!*

The approach to expand the WISC historical events and determine the frequencies of the offspring probabilistic storms (GEV distribution fitted to the historical SSI values) has merit, and the concluding results in paragraphs 3.2 and 3.3, also provided in table 2, are realistic. [...] I understand why the authors prefer to retain the confidence interval based on the WISC historical set (CHF 19M to 33,000M), yet this reduces somewhat the functionality of the probabilistic expansion model. It's main objective is to provide a tail view. Here are a few suggestions: (a) Sample randomly the equivalent of 250 or 500 years of storms and build multiple exceedance frequency curves for each sample. (b) Estimate multiple probabilistic extensions of the WISC historic event set with different initial assumptions … (c) combination of the above two ideas.

Author's response: We thank the referee for his suggestions. We have implemented all of them and discuss the results in the following. As a conclusion, we would still argue, that the yellow ribbon in Fig. 2

(i.e., the sampling uncertainty of the modelled damages based on "WISC 168 historic") is the best illustration of the uncertainty for "WISC probabilistic extension". We will include this argumentation in the manuscript, alongside the arguments already provided in this response. [...] We are aware that the parameter uncertainty regarding the event set "WISC probabilistic extension" is important, especially in comparison with "WISC historic". However, in our opinion this source of uncertainty is not fully estimated and sufficiently illustrated with 184 such a resampling methodology. [...] We resampled (choice with replacement) the historic events [...] Then we created a probabilistic event set for each of these samples. The 90-% confidence interval is again given by the $5^{th}$ and $95^{th}$ percentiles of all samples. This is the best possible way we achieved to illustrate at least part of the uncertainty that originates from the fact that the best-estimate of the distribution of the pan-European Storm Severity Index is unknown and thusly the parameters for the creation of the probabilistic sets can only be chosen with a certain degree of uncertainty. The uncertainty estimation up until a 30-year return period follows approximately the uncertainty estimation for "WISC historic"; at higher return periods the uncertainty estimation is levelling off, probably due to the limited ability of our probabilistic approach to create very different (e.g., much stronger) events from the seeding historic set. Therefore, we argue that the shown difference between the yellow ribbon and the red ribbon could be misleading. [...] The results for the referee's suggestion 2c, which is a combination of his suggestions 2a and 2b, are given in Fig. R1-3. [...] Whereas this combination provides a smooth illustration of the resampling uncertainty, it still suffers from the same problem as the illustration in Fig. R1-2. Therefore, we would still argue that the yellow ribbon in Fig. 2 is the best illustration of the uncertainty for "WISC probabilistic extension".

Reviewer's response: Thanks for the extensive work that resulted in the uncertainty estimations given by the red ribbons in Fig R1-1 to 3. I think that this analysis illustrates very clearly three different levels of uncertainty estimations, blue, yellow and red ribbons. I should clarify that I do agree with your conclusion, the yellow ribbon gives the best account of the uncertainties associated with the "WISC

probabilistic extension" and it should be included in the paper. This does not change my view though that the 'reduced' uncertainty in the red ribbon from the resampling approach also has merit and it should be included in the paper as well, not to replace the (yellow ribbon) full WISC probabilistic extension uncertainty, but to complement it. Yes, the uncertainty estimation from resampling is 'incomplete' yet it can be helpful in the practical case of model. The Uncertainty in the yellow ribbon is too broad to provide a comparison criterion between two different exceedance frequency curves from different models, (e.g.

WISC hazard + GVZ versus WISC hazard + CLIMADA) Thus, I think inclusion of the resampling uncertainty (red ribbon) in addition to the full WISC probabilistic extension uncertainty) yellow ribbon can be advantageous for your paper.

*The referee is right to point out the advantage of the red ribbon, illustrating the range of the*

*bootstrapped damage frequency curves of the modelled damages based on "WISC probabilistic*

*extension". Therefore, we would like to include the referee's suggestion in the revised*

*manuscript, specifically the importance of the tail view of probabilistic datasets and the*

*comparison of different damage models for certain applications in the insurance industry. In*

*order to show the uncertainty range of the individual damage datasets, we use the following*

*ribbons in Fig. 2 of the paper:*

*1.  Blue ribbon: sampling uncertainty of insured damages*

*2.  Yellow ribbon: sampling uncertainty of modelled damages based on "WISC historic"*

*3.  Red ribbon: probabilistic envelope of the modelled damages based on "WISC probabilistic*

*extension"*

*Please note that the red ribbon with "probabilistic envelope" is labelled differently than the other*

*two ribbons (i.e., sampling uncertainty), which is to emphasise that the red ribbon was derived*

*differently than the other two.*

*Specifically, we would make the following additions to the sections Data and methods, Results, and Discussion of the manuscript as well as add the red ribbon in Fig. 2 and change the caption accordingly. In addition, throughout the paper we try to specify more precisely which source of uncertainty is meant at the respective point in the text.*

1.  *Section 2. Data and methods (paragraph at the end of Sect. 2.3.4, Line 374):*

**"However, for certain applications in the insurance industry the tail view of "WISC probabilistic extension" is an important feature of the dataset. The sampling uncertainty of "WISC historic" is too large to provide, for instance, a comparison criterion between two different exceedance frequency curves from different models. Therefore, we propose to illustrate the probabilistic content of "WISC probabilistic extension" by using bootstrapping of all probabilistic damage events. In this way, a "probabilistic envelope" around the best-guess exceedance frequency curve can be determined (see also Sect. 2.4.2). This way of illustration shows how the problem could be addressed in practice, knowing well that it does not illustrate the full uncertainty. In contrast to the sampling uncertainty, the probabilistic envelope could represent something like the "represented uncertainty". In the approach applied, we firstly bootstrapped (random sampling with replacement, number of samples = 100) the historic damage events and then used these samples to create an ensemble of probabilistic damage event sets. Secondly, for each new probabilistic damage event set, we bootstrapped (number of samples = 20) randomly the equivalent of 500 years of windstorm events and built an exceedance frequency curve for each sample. From this set of double-bootstrapped damage event sets (total number of samples = 2000), we then calculated the span between the 5th percentile to the 95th percentile for each exceedance frequency to illustrate the envelope of the probabilistic content."**

2.  *Section 3. Results (paragraph at the end of Sect. 3.3, Line 426):*

**"The red ribbon in Fig. 2 shows a possibility to illustrate the probabilistic envelope for the modelled damages based on "WISC probabilistic extension" and the GVZ damage model, according to a bootstrapping approach as described in Sect. 2.4.3. As expected, the probabilistic envelope for "WISC probabilistic extension" is much smaller than the range of sampling uncertainty for "WISC historic" (yellow ribbon)."**

3.  *Section 4. Discussion (Line 519):*

**"The fact that the probabilistic envelope for the modelled damages based on "WISC**
**probabilistic extension" (red ribbon in Fig. 2) does not cover the full range of the**
**sampling uncertainty for the modelled damages based on "WISC historic" (yellow**
**ribbon) shows two things: on the one hand, it shows the tail view, which is possible with**
**the help of "WISC probabilistic extension" for certain applications in the insurance**
**industry for instance; on the other hand, it reveals the limitations of the statistical**
**perturbation, which is used in the generation of "WISC probabilistic extension", to fully**
**represent the sampling uncertainty of the underlying historic data. Despite this**
**mismatch, […]. In future studies, the information from dynamical models, which are**
**run for many model years, would help to further reduce the sampling uncertainty**
**compared to this study."**

*4.   Figure 2 (Line 795):*

*The red ribbon is added and the caption is changed accordingly.*

[Figure]

**"Figure 2: […] The red ribbon shows the probabilistic envelope for the modelled**

**damages based on "WISC probabilistic extension" and the GVZ damage model**

**computed by applying a bootstrapping approach as described in Sect. 2.4.3. […]"**

The remaining two discussion topics in the review (the role of the loss uncertainty due to the vulnerability and the different input exposures) has also been addressed thoroughly and I consider them clarified.

[revised manuscript text omitted]

However, for certain applications in the insurance industry the tail view of "WISC probabilistic extension" is an
important feature of the dataset. The sampling uncertainty of "WISC historic" is too large to provide, for
instance, a comparison criterion between two different exceedance frequency curves from different models.
Therefore, we propose to illustrate the probabilistic content of "WISC probabilistic extension" by using
bootstrapping of all probabilistic damage events. In this way, a "probabilistic envelope" around the best-guess
exceedance frequency curve can be determined (see also Sect. 2.4.2). This way of illustration shows how the
problem could be addressed in practice, knowing well that it does not illustrate the full uncertainty. In contrast
to the sampling uncertainty, the probabilistic envelope could represent something like the "represented
uncertainty". In the approach applied, we firstly bootstrapped (random sampling with replacement, number of
samples = 100) the historic damage events and then used these samples to create an ensemble of probabilistic
damage event sets. Secondly, for each new probabilistic damage event set, we bootstrapped (number of
samples = 20) randomly the equivalent of 500 years of windstorm events and built an exceedance frequency
curve for each sample. From this set of double-bootstrapped damage event sets (total number of
samples = 2000), we then calculated the span between the 5th percentile to the 95th percentile for each
exceedance frequency to illustrate the envelope of the probabilistic content.

**3 Results**

**3.1 Single events**

[revised manuscript text omitted]